# Beyond the Known: An Unknown-Aware Large Language Model for Open-Set Text Classification

**Xi Chen**[1,2], **Chuan Qin**[2,3]\*, **Ziqi Wang**[1], **Shasha Hu**[1], **Chao Wang**[4], **Hengshu Zhu**[2,3], **Hui Xiong**[5,6]\*

[1] School of Computer Science and Technology, University of Science and Technology of China
[2] Computer Network Information Center, Chinese Academy of Sciences
[3] University of Chinese Academy of Sciences
[4] School of Artificial Intelligence and Data Science, University of Science and Technology of China
[5] AI Thrust, The Hong Kong University of Science and Technology (Guangzhou)
[6] Department of Computer Science and Engineering,
The Hong Kong University of Science and Technology, Hong Kong SAR
{chenxi0401, wzq142857, shashahu}@mail.ustc.edu.cn,
{chuanqin0426, zhuhengshu}@gmail.com,
wangchaoai@ustc.edu.cn, xionghui@ust.hk

## Abstract

Open-set text classification (OSTC) requires models to correctly classify in-distribution (ID) samples while reliably rejecting out-of-distribution (OOD) inputs—an essential capability for real-world NLP systems. However, existing approaches largely follow the post-hoc OOD detection paradigm after a closed-world training, optimizing predictive distributions solely over known label spaces and thus producing overconfident and biased predictions on OOD inputs. In this work, we present UnLLM, an Unknown-aware Large Language Model for OSTC. Instead of fixing classification to the entire known label space, we reformulate it into a subset-conditioned classification task: the LLM is prompted with sampled subsets of known labels, and any instance outside the candidate set is explicitly assigned as an "unknown" class. This reformulation transforms OOD detection from a post-hoc procedure into an intrinsic modeling capability. Grounded in this formulation, the modeling of the "unknown" is further systematically realized through a unified **representation–probability–inference** optimization, which progressively strengthens the model's capacity to capture open-set risk. Extensive experiments across six benchmarks show that UnLLM consistently outperforms state-of-the-art (SOTA) baselines. Code and datasets are available at `https://github.com/cx9941/UnLLM`.

## 1 Introduction

Text classification is a cornerstone task in natural language processing (NLP), underpinning diverse applications such as topic categorization (Prakhya et al., 2017), document management (Shu et al., 2017), and intent recognition (Zhou et al., 2023; Qin et al., 2025b). Most conventional models, however, operate under a closed-world assumption (Fei & Liu, 2016), presuming that all classes encountered at inference time are known during training. This unrealistic assumption restricts the applicability of these models in real-world, dynamic scenarios where encountering OOD instances—is common. Consequently, the paradigm of OSTC (Prakhya et al., 2017) emerged, necessitating models capable of accurately classifying ID samples and concurrently identifying OOD samples.

A common approach is to first train a neural network on ID data and then apply post-hoc OOD detection techniques such as MSP (Hendrycks & Gimpel, 2016) and OpenMax (Bendale & Boult, 2016). More recent efforts fine-tune pretrained language models (PLMs) to establish compact decision boundaries through contrastive and prototype-based learning, as in ADB (Zhang et al., 2021),

---
\*Corresponding Authors

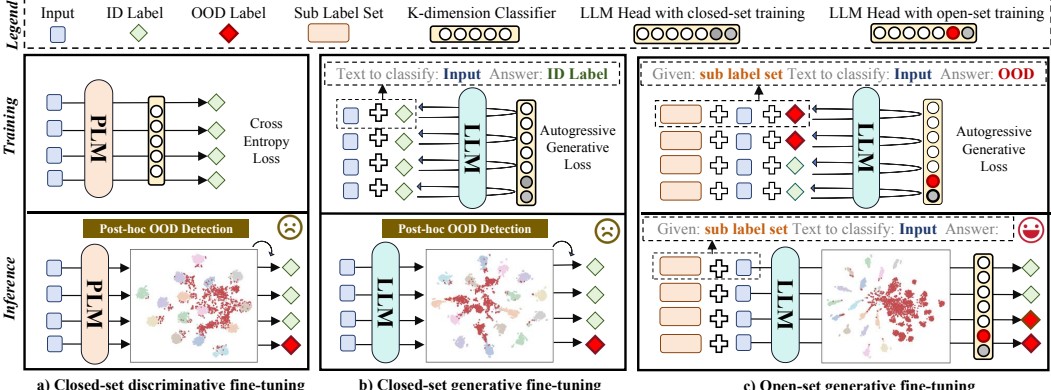

Figure 1: Illustration of different fine-tuning methods. Representation visualizations show significant overlap between ID and OOD (red) in a) and b), whereas c) demonstrates clear separation.

KNNCon (Zhou et al., 2022), and CLAP (Liu et al., 2023). However, these PLM-based methods following a discriminative paradigm often produce narrowly concentrated embeddings that hinder OOD detection. To address this issue, LLM-OOD (Liu et al., 2024) introduced a generative fine-tuning framework that reformulates classification as text generation and leverages final-token representations, which yield more distinguishable embeddings, improving OOD separability.

Nevertheless, both discriminative and generative fine-tuning remain confined to the *closed-set* paradigm, where models are optimized exclusively on ID labels and lack explicit open-set supervision (Scheirer et al., 2012). In computer vision, several methods (e.g., VOS (Du et al., 2022), NPO (Tao et al., 2023)) attempt to synthesize virtual outliers to regularize decision boundaries. However, these synthetic outliers cannot be guaranteed to represent true OOD instances—particularly under sparse ID coverage—introducing label noise and constraining generalization. This limitation raises a critical question: *Is genuine open-set training—where OOD detection is integrated with* guaranteed-correct *supervision—fundamentally unattainable?*

To explore this question, we first conduct systematic analysis and visualization of the differences between the two existing paradigms—discriminative and generative fine-tuning—as shown in Figure 1.a and 1.b. This analysis yields two key insights. First, the generative fine-tuning paradigm inherently benefits from a broader output space in the LLM's classification head. This expanded dimensionality, combined with the extensive general knowledge acquired during pretraining, enables the model to learn more generalized and contextually rich representations, thereby improving its resilience to open-set risk. However, generative fine-tuning still assumes that the label space is identical to the entire ID label set, restricting the model to optimize final-token representations only over ID label tokens. The gap between label spaces at training and testing further degrades predictive performance, raising a natural idea: *If we explicitly model this gap during training, can we effectively eliminate such bias?* Motivated by this, **we are the first to propose an open-set training paradigm tailored for LLMs**. Specifically, by conditioning the LLM on partial subsets of labels, we construct conditional-OOD samples—instances whose ground-truth labels are deliberately excluded from the candidate label set. This formulation enables the model to explicitly perceive open-set risk during training. Unlike prior methods that treat OOD detection as a post-hoc decision process, our approach opens the LLM head to directly optimize parameters associated with OOD-relevant tokens. This design allows the model to go beyond implicit uncertainty estimation and explicitly model the semantic space of the unknown, thereby enhancing its capacity to recognize OOD inputs.

However, this design still faces several key challenges in practice: (1) Distribution gap between conditional-OOD and real OOD samples: Conditional-OOD samples are constructed from known classes with clear semantic boundaries. In contrast, real OOD samples often lie near the decision boundaries between multiple known classes. (2) Misalignment between internal knowledge and output: Even though "*open-set training*" can separate ID and OOD samples at the representation level, standard generative strategies may fail to faithfully reflect the model's internal knowledge. (3) Overconfidence in label-similar OODs: LLMs often assign overly confident predictions to OOD inputs that are semantically close to known labels, leading to erroneous decisions.

To overcome these, we propose UnLLM based on open-set training paradigm that strengthens the modeling of the unknown across three tightly connected levels: (1) **Representation modeling**, where open-set generative fine-tuning, combined with contrastive learning and orthogonality constraints, produces compact ID embeddings and well-separated known/unknown representations, thereby reducing the distribution gap; (2) **Probability calibration**, which leverages OOD parameter calibration to align internal activations with unknown semantics, ensuring consistency between internal knowledge and probabilistic outputs; and (3) **Reflective inference**, which employs analogy-augmented self-reflection during inference to mitigate overconfidence on semantically confusing OOD cases. Comprehensive experiments conducted across six benchmark datasets demonstrate that our proposed UnLLM, consistently surpasses SOTA baselines. Additional visualization analyses further substantiate our model's capability to distinctly separate ID and OOD representations, successfully addressing the persistent bottleneck posed by "*closed-set training*" paradigms.

## 2 RELATED WORKS

**Open-Set Text Classification**  Early OSTC methods relied on maximum softmax probability (MSP)(Hendrycks & Gimpel, 2016) to reject uncertain predictions. However, researchers (Bendale & Boult, 2016) argued that MSP merely rejects uncertain predictions, OpenMax(Bendale & Boult, 2016) improved over MSP by fitting a Weibull distribution to logits. DOC (Shu et al., 2017) introduced one-vs-rest sigmoid classifiers to reduce open space risk, while DeepUnk (Lin & Xu, 2019) leveraged large-margin losses for better separation. Nevertheless, they still lack explicit decision boundaries for unknown classes. Recent works exploit PLMs to refine decision boundaries. ADB (Zhang et al., 2021) proposed adaptive spherical margins, CLAP (Liu et al., 2023) optimized boundary scaling, and KNNCon (Zhou et al., 2022) combined contrastive training with local outlier detection. Researchers (Zhou et al., 2023) noted that PLMs often "overthink" semantic features, leading to overfitting on ID labels. To address this, they proposed DyEn, which employs confidence-based early exits to mitigate overfitting. However, the overthinking problem remains fundamentally unresolved, as it is not explicitly optimized during training.

**OOD Detection**  OOD detection is closely tied to OSTC and has been widely studied. Confidence-based methods such as MSP (Hendrycks & Gimpel, 2016), OpenMax (Bendale & Boult, 2016), and Energy (Liu et al., 2020) dominated early research, with extensions incorporating auxiliary OOD data (Mohseni et al., 2020)—often impractical in real settings. To avoid external supervision, ODIN (Hsu et al., 2020) introduced input perturbations, while VOS (Du et al., 2022) and NPO (Tao et al., 2023) synthesized virtual outliers to regularize decision boundaries. However, such synthetic samples poorly approximate real unknowns, especially under sparse ID coverage, resulting in limited generalization. In recent years, LLMs have advanced rapidly, demonstrating remarkable capabilities (Qin et al., 2025a) and achieving strong performance across a wide range of downstream tasks (Chen et al., 2023; Tong et al., 2025). LLM-OOD (Liu et al., 2024) demonstrated that discriminative PLMs suffer from low-isotropy embeddings that hinder OOD separation. They proposed a generative fine-tuning paradigm, reformulating classification as text generation and exploiting token-level representations, which improved embedding isotropy and OOD detection performance. VALID (Emde et al., 2025) also pointed out that LLMs often lack explicit awareness of domain boundaries and used likelihood-ratio-based OOD detection to restrict out-of-domain generations.

Nonetheless, both discriminative and generative approaches still rely on closed-set training with post-hoc detection, leaving models fundamentally unaware of unknown classes during training.

## 3 PRELIMINARIES

**Problem Definition**  Following the open-set learning setting (Scheirer et al., 2012), the training set $\{(x, y)\}_i^N$ with $N$ textual input instances are drawn from a product space $\mathcal{D}^l = \mathcal{X} \times \mathcal{Y}^l$, where $\mathcal{X}$ is the input space, and $\mathcal{Y}^l = \{1, \cdots, K\}$ is the label space of known classes. During the test period, some samples might belong to none of the known classes, which can be allocated to one super unknown class $K + 1$. These OOD samples are drawn from a product space $\mathcal{D}^u = \mathcal{X} \times \mathcal{Y}^u$, where $\mathcal{Y}^u = \{K + 1\}$ is the label space of unknown classes. OSTC aims to classify ID inputs into one of the known classes $\mathcal{Y}^l$ and identify OOD inputs by assigning them to the unknown classes $\mathcal{Y}^u$.

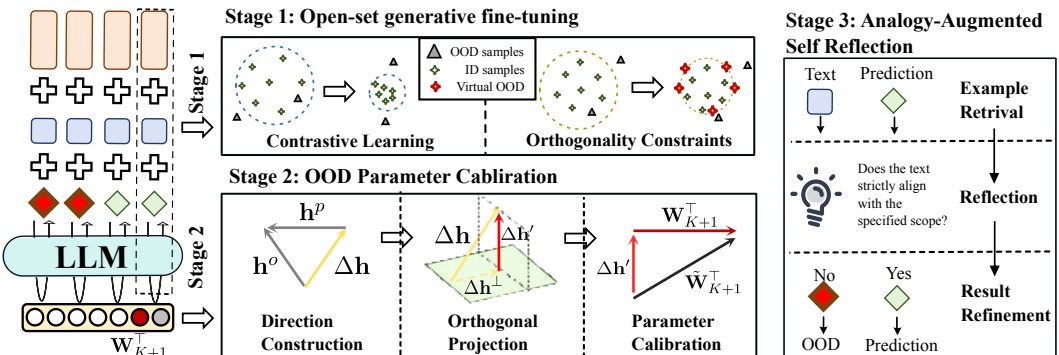

Figure 2: An illustration of our proposed UnLLM.

**Generative Fine-tuning with LLMs**  We outline the generative fine-tuning used in LLM-OOD (Liu et al., 2024), which leverages the inherent generative capabilities of LLMs for text classification. This approach formulates the classification task as a conditional text generation problem. Specifically, given an input $x$, we expand it using a template to streamline output extraction, such as: "## Input: $x$. ## Output: $y$". Subsequently, we maximize the likelihood of generating the label tokens y: $\max \mathbf{P}(y \mid x) = \prod_{i=1}^{|y|} \mathbf{P}_\theta(y_i \mid x, y_{<i})$, where $\theta$ represents the parameters, $|y|$ is the length of the label in tokens, and $y_{<i}$ refers to the tokens preceding the current prediction token $y_i$.

## 4 METHOD

As illustrated in Figure 2, UnLLM comprises three stages. **Stage 1: Open-Set Generative Fine-tuning**: We employ a label-partitioning strategy to generate both ID and pseudo-OOD inputs for generative fine-tuning, allowing the model to develop a K+1-class classifier without relying on external data. This stage integrates contrastive learning to enhance intra-class representation compactness. Additionally, we synthesize a virtual OOD representation plane and apply an orthogonality constraint loss to enforce a clear separation between ID and OOD representations. **Stage 2: OOD Parameter Calibration**: We calibrate the parameter of the K+1-class tokens in the LLM's Head guided by validation-set bias. This ensures alignment between the model's internal representations and its external outputs. **Stage 3: Analogy-Augmented Self-Reflection**: To mitigate overconfidence in ambiguous cases, we retrieve relevant examples in the training set during inference to facilitate analogical reasoning, enabling the model to verify whether a given text is OOD.

### 4.1 OPEN-SET GENERATIVE FINE-TUNING

Traditional methods often suffer the vulnerability to OOD inputs due to the lack of explicit knowledge of unknowns during training time (Du et al., 2022). To allow LLM to perceive the task of OOD detection, we reformulate the training objective from $\max \mathbf{P}(y|x)$ to $\max \mathbf{P}(\tilde{y}|x, \mathcal{Y}^p)$, where $\mathcal{Y}^p$ represents a subset of the $\mathcal{Y}^k$ and $\tilde{y}$ is a partition-conditional label defined as $\tilde{y} = y$ if $y \in \mathcal{Y}^p$ and $\tilde{y} = K + 1$ otherwise. This formulation constructs OOD samples labeled as $\tilde{y} = K + 1$, allowing the model to recognize the boundaries of each label and mitigate overfitting on OOD inputs.

**Label Partition**  We introduce a label partition strategy to construct sub-label sets. Specifically, for a given ID training instance $x_i$, the label set is evenly divided into $s$ partitions, $\{\mathcal{Y}_i^p \mid i \in [1, s]\}$, ensuring that labels in each partition are mutually exclusive. For each label partition $\mathcal{Y}_{i,j}^p \in \mathcal{Y}_i^p$, the corresponding partition-conditional label is denoted as $\tilde{y}_{i,j}$. Let $c_1, c_2, \dots$ represent the candidate labels within $\mathcal{Y}_{i,j}^p$. We construct the following textual input for the model:

---

**You are an expert in text classification.**
*Text to classify: $x_i$.*
*Given candidate categories:* [NID($c_1$).Text($c_1$), NID($c_2$).Text($c_1$), . . . , NID($K + 1$).Text($K + 1$)].
*Which category does the text belong to?*
**Answer:** NID($\tilde{y}_{i,j}$).Text($\tilde{y}_{i,j}$)

---

Here, Text($k$) is the textual content of label $k$ and NID($\cdot$) is a numbering method for consistent label formatting, as detailed in Appendix A.1.

**Generative Fine-tuning**   We follow the generative fine-tuning (Liu et al., 2024), with the objective to maximize the likelihood of the label $\tilde{y}_{i,j}$:

$$\mathcal{L}_{\text{gen}} = \sum_{i,j} \sum_{k=1}^{|\tilde{y}_{i,j}|} \log \mathbf{P}_\theta \left( \tilde{y}_{i,j,k} \mid x_i, \mathcal{Y}_{i,j}^p, \tilde{y}_{i,j,<k} \right), \tag{1}$$

where $\tilde{y}_{i,j}$ denotes the tokens of NID($\tilde{y}_{i,j}$), $|\tilde{y}_{i,j}|$ is the length of the token sequence, and $y_{i,j,<k}$ refers to the tokens preceding the current token $y_{i,j,k}$.

**Contrastive Learning**   To further model discriminative representations, we utilize contrastive learning (Zeng et al., 2021) to maximize inter-class variance and minimize intra-class variance. Specifically, we extract the representations of the $\tilde{y}_{i,j}$ tokens at the last layer of LLM as $\mathbf{h}_{i,j} \in \mathbb{R}^{t \times d}$, where $t$ is the length of $\tilde{y}_{i,j}$ token sequence and $d$ is the embedding size of LLM. We normalize the $\mathbf{h}_{i,j}$ as $\tilde{\mathbf{h}}_{i,j}$ and define the contrastive loss with $N_{\tilde{y}_{i,j}}$ as the number of examples of $\tilde{y}_{i,j}$:

$$\mathcal{L}_{cl} = \sum_{i,j} -\frac{1}{N_{\tilde{y}_{i,j}} - 1} \sum_{(i,j) \neq (i',j'), \tilde{y}_{i,j} = \tilde{y}_{i',j'}} \log \frac{\exp(\tilde{\mathbf{h}}_{i,j} \cdot \tilde{\mathbf{h}}_{i',j'})}{\sum_{(i,j) \neq (i'',j'')} \exp(\tilde{\mathbf{h}}_{i,j} \cdot \tilde{\mathbf{h}}_{i'',j''})}. \tag{2}$$

**Orthogonality Constraints**   To sharpen the decision boundaries of ID labels, we construct a virtual OOD subspace by sampling outliers from low-likelihood regions of each class distribution (Du et al., 2022; Tao et al., 2023). Unlike VOS and DPO, which directly regularize on such synthetic samples, we note their inherent estimation bias and limited fidelity to real OOD data. Drawing inspiration from ViM (Wang et al., 2022), we project virtual OOD features onto the principal component plane and enforce orthogonality with the ID subspace. This orthogonalization emphasizes the salient structure of the OOD subspace while suppressing noise.

Specifically, we assume that the representation for each class $k \in [1, K]$ follows a class-conditional Gaussian distribution: $\mathbf{P}_\theta(\mathbf{h}_{i,j} \mid y_{i,j} = k) \sim \mathcal{N}(\mu_k, \sigma_k^2)$, where $\mu_k \in \mathbb{R}^{t \times d}$ and $\sigma_k \in \mathbb{R}^{t \times d}$. To estimate $\mu_k$ and $\sigma_k$, we maintain a class-conditional queue $\mathcal{Q}_k$. The empirical class mean $\widehat{\mu}_k$ and variance $\widehat{\sigma}_k^2$ are calculated as: $\widehat{\mu}_k = \frac{1}{|\mathcal{Q}_k|} \sum_{\mathbf{h}_{i,j} \in \mathcal{Q}_k} \mathbf{h}_{i,j}$, $\widehat{\sigma}_k^2 = \frac{1}{|\mathcal{Q}_k|} \sum_{\mathbf{h}_{i,j} \in \mathcal{Q}_k} (\mathbf{h}_{i,j} - \widehat{\mu}_k)^2$.

For each $k$, we sample virtual outliers $\mathbf{v}_k \in \mathbb{R}^{t \times d}$ from the $\epsilon$-likelihood region of the estimated class-conditional distribution: $\frac{1}{\sqrt{2\pi\widehat{\sigma}_k^2}} \exp\left(-\frac{(\mathbf{v}_k - \widehat{\mu}_k^2)}{2\widehat{\sigma}_k^2}\right) < \epsilon$, where $\epsilon$ is a dynamic value chosen based on the smallest likelihood in $\mathcal{Q}_k$, ensuring that the sampled outliers are located near the boundaries.

The sampled virtual outliers are aggregated to approximate the critical regions near decision boundaries. Rather than imposing pairwise constraints, we generalize them into a principal subspace $\mathbf{O} \in \mathbb{R}^{td \times e}$ obtained via principal component analysis (PCA), where $e$ is the number of critical characteristics. ID features are flattened into $\mathbf{H}_{\text{ID}} \in \mathbb{R}^{N_{\text{ID}} \times td}$, where $N_{\text{ID}}$ is the number of ID samples in the batch. To promote better separation, we minimize $\mathcal{L}_{\text{orth}}$ to ensure the orthogonality:

$$\mathcal{L}_{\text{orth}} = \|\mathbf{H}_{\text{ID}}\mathbf{O}\|_F^2, \tag{3}$$

where $\|\cdot\|_F^2$ denotes the Frobenius norm. The details of PCA are put in Appendix A.2.

**Learning Objectives**   We jointly fine-tune the LLM with a composite loss: $\mathcal{L} = \lambda_{\text{cl}}\mathcal{L}_{\text{cl}} + \lambda_{\text{orth}}\mathcal{L}_{\text{orth}} + \mathcal{L}_{\text{gen}}$, where $\lambda_{\text{cl}}$ and $\lambda_{\text{orth}}$ are the hyperparameters. By jointly optimizing $\mathcal{L}$, the LLM achieves compact ID intra-class representations while maintaining distinct OOD boundaries.

## 4.2   OOD PARAMETER CALIBRATION

After training, the LLM gains an awareness of the OOD detection task and can effectively distinguish between ID and OOD representations. However, we observe a misalignment between the LLM's internal knowledge (representation space) and its outputs. This discrepancy arises due to the standard generation method, which relies on token-level probabilities and fails to provide meaningful OOD confidence estimates (Kapoor et al., 2024). Previous studies suggest that the activation

space of many LLMs contains interpretable directions (Li et al., 2024). Motivated by this, we propose identifying a calibration direction in the activation space and calibrating the OOD token weight, $\mathbf{W}_{K+1} \in \mathbb{R}^{d \times t}$, in the LLM's output layer to produce calibrated probabilities for $K + 1$ class.

**Calibration Direction Construction**  First, we evaluate the fine-tuned LLM on the label-partitioned validation set to identify false ID samples $\mathcal{X}^p$, false OOD samples $\mathcal{X}^o$, and correctly predicted samples $\mathcal{X}^r$. The corresponding representations are extracted as $\mathbf{H}^p \in \mathbb{R}^{|\mathcal{X}^p| \times t \times d}$, $\mathbf{H}^o \in \mathbb{R}^{|\mathcal{X}^o| \times t \times d}$, and $\mathbf{H}^r \in \mathbb{R}^{|\mathcal{X}^r| \times t \times d}$. Next, we derive the average representative vectors for false positives, false negatives, and correct predictions, denoted as $\mathbf{h}^p \in \mathbb{R}^{t \times d}$, $\mathbf{h}^o \in \mathbb{R}^{t \times d}$ and $\mathbf{h}^r \in \mathbb{R}^{t \times d}$. To ensure the OOD token weight of the LLM head $\mathbf{W}_{K+1}^\top \in \mathbb{R}^{t \times d}$ aligns with OOD samples while being distant from ID samples, we define the calibration direction as $\Delta\mathbf{h} = \mathbf{h}^p - \mathbf{h}^o \in \mathbb{R}^{t \times d}$ which indicates how $\mathbf{W}_{K+1}^\top$ should be adjusted.

**Orthogonal Projection**  To ensure that adjustments do not disrupt correct predictions, we project $\Delta\mathbf{h}$ onto the orthogonal subspace of the representations derived from true predictions $\mathbf{h}^r$. This orthogonal projection problem has the following closed-form solution (Zhang et al., 2018) as:

$$\Delta\mathbf{h}^\perp = (\mathbf{h}^r(\mathbf{h}^{r\top}\mathbf{h}^r)^{-1}\mathbf{h}^{r\top})\Delta\mathbf{h}. \tag{4}$$

To isolate the OOD-specific adjustment direction while preserving the model's ability to predict correctly, we subtract $\Delta\mathbf{h}^\perp$ from $\Delta\mathbf{h}$, obtaining the calibration vector: $\Delta\mathbf{h}' = \Delta\mathbf{h} - \Delta\mathbf{h}^\perp \in \mathbb{R}^{t \times d}$.

**Parameter Calibration**  Finally, we calibrate $\mathbf{W}_{K+1}^\top$ by shifting it in the $\Delta\mathbf{h}'$ direction: $\tilde{\mathbf{W}}_{K+1}^\top = \mathbf{W}_{K+1}^\top + \lambda_v \Delta\mathbf{h}'$, where $\lambda_v$ is a hyperparameter controlling the calibration magnitude. By calibrating $\mathbf{W}_{K+1}^\top$, which represents the OOD mapping function, this approach aligns the model's internal knowledge for OOD detection, without additional training.

### 4.3 ANALOGY-AUGMENTED SELF-REFLECTION

During the inference phase, the fine-tuned LLM processes test samples using the same label partition strategies. For each test sample, the model sequentially evaluates it against label subsets, stopping once an ID label is identified. If no ID label is found, the sample is classified as OOD. However, we observe that LLMs often exhibit overconfidence in their predictions, especially for texts with high semantic similarity to known labels.

Inspired by analogy-augmented generation (Roth et al., 2024), a strategy that leverages past experiences to address unfamiliar problems, we propose an analogy-augmented self-reflection approach to address this challenge. Specifically, given an instance $x_i$ and its generated label $\hat{y}_{i,j}$, we retrieve examples most similar to $x_i$ based on embeddings computed by a PLM associated with $\hat{y}_{i,j}$. These retrieved examples, denoted as $\{a_1, a_2, \ldots\}$, are ranked by similarity as follows:

$$\text{Sim}(x_i, a_1) \geq \text{Sim}(x_i, a_2) \geq \ldots, \ \text{Sim}(x_i, a_j) = \text{Cos}(\text{LM}(x_i), \text{LM}(a_j)), \tag{5}$$

where $\text{Cos}(\cdot, \cdot)$ denotes the cosine similarity function and $\text{LM}(\cdot)$ represents the embedding function.

The retrieved analogical examples are then feed to the LLM for further reflection as follows:

> *Recall relevant exemplars of $\hat{y}_{i,j}$: $a_1, a_2, \cdots$.*
> *Does the text strictly align with the specified scope? Please start by answering Yes or No.*
> **Answer:**

Samples receiving a "No" response are classified as OOD. This analogy-enhanced reflection allows the LLM to better understand label semantics, reducing biases from superficial semantic similarity.

## 5 EXPERIMENT

### 5.1 EXPERIMENTAL SETTINGS

**Datasets** We evaluated our method on six open-set text classification benchmark datasets: BANK-ING (Casanueva et al., 2020), CLINC (Larson et al., 2019), StackOverflow (Xu et al., 2015), News-

groups (Schneider, 2003), Reviews (Jindal & Liu, 2008), and THUCNews (Li et al., 2006). In particular, THUCNews is a Chinese dataset, while the others are English datasets and was randomly sampled to match the size of other datasets. Detailed descriptions are put in Appendix B.1.

**Dataset Split** To ensure fair comparison with prior methods, we aligned dataset splits as closely as possible with existing work. For the BANKING, CLINC, and StackOverflow benchmarks, we directly used the provided splits in previous studies (Zhou et al., 2023). For Newsgroups and Reviews, where pre-defined splits are unavailable (Shu et al., 2017), we followed the methodology in prior work to divide the data; the same procedure was applied to THUCNews. Consistent with (Zhou et al., 2023), we retained 25%, 50%, or 75% of the classes as ID, while the remaining classes were treated as OOD. OOD samples were excluded from training and validation, and reserved for testing. Detailed dataset statistics are reported in Appendix B.2.

**Baselines** We compared UnLLM against mainstream OSTC methods, which have been carefully discussed in Section 2, including discrimitive training-based baselines: DOC (Shu et al., 2017), DeepUnk (Lin & Xu, 2019), ADB (Zhang et al., 2021), CLAP (Liu et al., 2023), KnnCon (Zhou et al., 2022), and DyEn (Zhou et al., 2023). To the best of our knowledge, no prior work has systematically applied LLMs to OSTC. For a fair comparison, we construct LLM-based baselines by fine-tuning LLMs on the ID training set and adapting existing OOD detection methods to this setting. For generative fine-tuning, we employ LLM-OOD (Liu et al., 2024). For discriminative fine-tuning, we adapt Energy-based scoring (Liu et al., 2020), VOS (Du et al., 2022), and NPO (Tao et al., 2023), where VOS and NPO leverage virtual OOD distributions to improve detection. More baseline implement details and OOD detection baselines, which perform less competitively, are provided in Appendix C.6.

**Evaluation Metrics** Following the evaluation approach in prior studies (Liu et al., 2023; Zhou et al., 2023), we treat all OOD classes as the K+1 class. K-F1 and N-F1 represent the macro F1-scores for ID and OOD classes, respectively, effectively capturing the model's performance on ID classification and OOD detection. Aligning with previous works, we also compute F1 and ACC, provided in Appendix C.6 due to page limitations.

## 5.2 OVERALL PERFORMANCE

Table 1 presents the performance comparison of our method against baselines across six datasets. The results clearly indicate that UnLLM achieves substantial and consistent improvements over SOTA baselines, demonstrating its effectiveness. Specifically, our method improves the K-F1 scores across all datasets by an average of 4.40%, 2.80%, and 2.55% and the N-F1 scores by 1.63%, 1.53%, and 5.09% under 25%, 50%, and 75% known class settings, respectively.

Several key insights can be drawn from the results. First, while UnLLM achieves the best overall performance, LLM-based baselines frequently rank second, underscoring the strong potential of large language models in OOD detection. However, it is noteworthy that generative training approaches (e.g., LLM-OOD) do not outperform discriminative counterparts (e.g., EnergyBased), suggesting that prior generative strategies fall short in capturing discriminative decision boundaries. In contrast, our conditional subset training paradigm enriches sample diversity during optimization, significantly boosting both classification accuracy and OOD detection ability.

Second, the results on Reviews and Newsgroups reveal that BERT-based methods struggle with long-text classification, whereas generative LLMs are more adept at modeling complex semantic structures. This semantic advantage contributes to stronger classification and OOD detection capabilities. Nevertheless, LLM-based approaches remain vulnerable to overfitting, as they rely on the conventional K-class classification paradigm. Our method addresses this limitation by explicitly modeling unknown-class awareness during training through a K+1 classification framework. This design effectively exploits the representational power of LLMs while aligning training and inference objectives, yielding both significant and stable performance improvements.

Finally, performance variations across different known class ratios reveal a consistent trend: as the ratio of known classes increases, K-F1 improves due to the greater availability of ID labels, facilitating more accurate ID classification. However, N-F1 generally declines, reflecting increased challenges in OOD detection. This decline occurs because models tend to overfit known labels as more labeled data becomes available. Notably, our approach maintains strong OOD detection

Table 1: Performance of various methods across 6 datasets at different ratios. Metrics include K-F1 and N-F1. The best results are highlighted in bold, while the second-best results are underscored. Each result represents the mean value of four repetitive experiments.

| Ratio | Backbone | Method | BANKING | | CLINC | | StackOverflow | | Reviews | | Newsgroups | | THUCnews | |
|---|---|---|---|---|---|---|---|---|---|---|---|---|---|---|
| | | | K-F1 | N-F1 | K-F1 | N-F1 | K-F1 | N-F1 | K-F1 | N-F1 | K-F1 | N-F1 | K-F1 | N-F1 |
| 0.25 | CNN | DOC | 66.40 | 81.57 | 75.46 | 85.63 | 69.81 | 83.61 | 69.81 | 79.57 | 66.21 | 89.59 | 56.43 | 82.27 |
| | LSTM | DeepUnk | 70.87 | 87.74 | 75.31 | 91.36 | 67.10 | 88.93 | 45.18 | 51.59 | 56.19 | 71.54 | 41.40 | 46.72 |
| | BERT | ADB | 51.07 | 61.84 | 59.00 | 66.67 | 75.91 | 88.37 | 48.27 | 68.56 | 28.70 | 34.02 | 58.98 | 62.20 |
| | | CLAP | 58.69 | 68.21 | 61.88 | 72.75 | 69.75 | 81.81 | 54.45 | 78.20 | 35.79 | 47.22 | 39.23 | 49.17 |
| | | KNNCon | 65.97 | 70.63 | 79.53 | 87.85 | 66.83 | 75.63 | 53.75 | 71.26 | 55.71 | 60.18 | 47.91 | 42.33 |
| | | DyEn | 65.53 | 73.71 | 77.16 | 86.85 | 60.17 | 55.88 | 44.28 | 42.11 | 57.74 | 67.04 | 43.99 | 17.90 |
| | LLaMA3.1-8B | LLM-OOD | 65.62 | 79.91 | 66.65 | 67.89 | 78.54 | 90.67 | 60.09 | 80.68 | 59.33 | 76.93 | 64.17 | 80.08 |
| | | EnergyBased | 71.07 | 90.82 | 83.10 | 91.46 | 80.91 | 94.57 | 48.21 | 89.69 | 60.45 | 83.35 | 61.72 | 92.41 |
| | | VOS | 74.39 | 89.82 | 61.79 | 90.95 | 62.06 | 89.64 | 61.35 | 89.09 | 61.28 | 84.66 | 54.59 | 89.72 |
| | | NPO | 73.83 | 89.73 | 82.60 | 90.29 | 86.01 | 95.60 | 54.25 | 89.78 | 59.19 | 85.49 | 61.06 | 92.79 |
| | | UnLLM | 75.04 | 92.02 | 83.90 | 93.58 | 88.65 | 96.00 | 62.16 | 91.94 | 68.33 | 91.82 | 83.63 | 94.46 |
| 0.5 | CNN | DOC | 74.50 | 77.92 | 84.10 | 84.38 | 77.96 | 79.57 | 55.42 | 67.52 | 77.86 | 74.67 | 46.63 | 64.58 |
| | LSTM | DeepUnk | 59.75 | 74.91 | 69.80 | 81.54 | 76.79 | 81.86 | 41.86 | 20.12 | 69.07 | 59.93 | 62.55 | 24.58 |
| | BERT | ADB | 63.09 | 56.92 | 71.14 | 61.67 | 83.95 | 83.41 | 51.56 | 49.18 | 33.99 | 22.00 | 76.67 | 69.88 |
| | | CLAP | 56.04 | 54.13 | 67.15 | 63.94 | 84.99 | 85.36 | 55.00 | 55.79 | 50.37 | 41.19 | 71.49 | 62.51 |
| | | KNNCon | 80.35 | 75.13 | 82.71 | 62.30 | 85.11 | 83.12 | 61.31 | 46.13 | 77.60 | 63.58 | 76.44 | 48.49 |
| | | DyEn | 78.55 | 68.34 | 88.38 | 82.46 | 77.06 | 60.25 | 60.47 | 27.64 | 77.83 | 62.19 | 71.79 | 34.68 |
| | LLaMA3.1-8B | LLM-OOD | 75.06 | 81.65 | 78.85 | 46.07 | 89.71 | 89.54 | 65.55 | 59.18 | 77.11 | 72.95 | 83.34 | 78.66 |
| | | EnergyBased | 74.96 | 82.76 | 88.39 | 90.50 | 89.58 | 90.57 | 47.32 | 74.93 | 75.39 | 76.77 | 80.66 | 80.41 |
| | | VOS | 78.93 | 83.94 | 66.38 | 80.99 | 58.88 | 76.33 | 61.71 | 77.04 | 76.88 | 74.99 | 77.78 | 80.22 |
| | | NPO | 78.70 | 84.12 | 91.31 | 91.53 | 90.14 | 90.78 | 58.23 | 76.82 | 78.17 | 76.73 | 82.37 | 79.90 |
| | | UnLLM | 82.74 | 85.72 | 93.42 | 93.00 | 91.91 | 92.60 | 68.81 | 80.59 | 83.95 | 77.20 | 84.83 | 80.75 |
| 0.75 | CNN | DOC | 78.85 | 61.25 | 87.37 | 72.81 | 84.90 | 72.30 | 53.95 | 52.53 | 67.67 | 57.58 | 29.77 | 41.32 |
| | LSTM | DeepUnk | 59.27 | 54.85 | 58.57 | 67.62 | 82.57 | 68.93 | 27.31 | 11.81 | 65.95 | 40.93 | 70.76 | 26.25 |
| | BERT | ADB | 72.07 | 43.48 | 78.40 | 51.69 | 87.65 | 75.10 | 47.35 | 35.55 | 35.07 | 17.93 | 79.59 | 54.59 |
| | | CLAP | 62.48 | 40.37 | 71.19 | 47.04 | 88.23 | 75.98 | 53.00 | 42.43 | 46.16 | 20.47 | 72.82 | 50.93 |
| | | KNNCon | 86.90 | 61.61 | 93.91 | 80.67 | 89.38 | 74.55 | 65.22 | 49.02 | 86.36 | 62.53 | 90.49 | 63.10 |
| | | DyEn | 86.79 | 54.74 | 93.38 | 78.02 | 85.43 | 54.09 | 64.24 | 21.81 | 85.24 | 47.06 | 87.88 | 28.73 |
| | LLaMA3.1-8B | LLM-OOD | 72.10 | 62.34 | 91.96 | 60.82 | 88.54 | 77.79 | 65.02 | 38.54 | 83.29 | 67.06 | 89.41 | 71.94 |
| | | EnergyBased | 76.84 | 63.80 | 90.18 | 79.11 | 85.94 | 74.07 | 30.08 | 44.55 | 70.78 | 56.69 | 79.41 | 70.34 |
| | | VOS | 79.68 | 65.65 | 84.64 | 72.68 | 83.54 | 67.85 | 48.83 | 49.18 | 75.72 | 59.23 | 83.85 | 74.05 |
| | | NPO | 79.89 | 65.79 | 93.24 | 83.14 | 90.10 | 79.22 | 39.08 | 46.74 | 76.68 | 60.36 | 78.67 | 70.35 |
| | | UnLLM | 88.58 | 72.60 | 96.94 | 89.73 | 92.36 | 83.08 | 67.84 | 56.65 | 89.06 | 72.18 | 93.55 | 78.11 |

performance even under highly known-class settings, demonstrating its ability to mitigate overfitting and maintain stable performance across diverse scenarios.

## 5.3 RESULTS ON UNLLM VS. LLMS WITH DIFFERENT PROMPTING STRATEGIES

We further compared UnLLM with LLMs that were not fine-tuned, under various prompting strategies including zero-shot, few-shot, chain-of-thought (CoT), and analogy-augmented self-reflection (Analogy) in Section 4.3. In particular, we evaluated three different backbones, namely LLaMA3.1-8B, Qwen2.5-32B, and DeepSeek-V3-0324, where DeepSeek-V3-0324 is one of SOTA open-source LLMs.

As shown in Table 2, across all ratios, in-context prompting methods (Zero/Few/CoT) yield relatively unstable performance. Few-shot even collapses under certain backbones (e.g., Qwen2.5-32B at 0.25 and 0.5 ratios), whereas CoT sometimes provides modest improvements but remains inconsistent. Analogy

Table 2: Performance comparison on CLINC dataset across different prompting strategies.

| Ratio | Method | LLaMA3.1-8B | | Qwen2.5-32B | | DeepSeek-V3-0324 | |
|---|---|---|---|---|---|---|---|
| | | K-F1 | N-F1 | K-F1 | N-F1 | K-F1 | N-F1 |
| 0.25 | Zero-Shot | 51.85 | 59.37 | 57.65 | 62.20 | 58.70 | 61.90 |
| | Few-shot | 49.45 | 63.01 | 17.52 | 84.30 | 58.81 | 57.87 |
| | Cot | 55.00 | 54.43 | 56.86 | 56.30 | 61.03 | 65.54 |
| | Analogy | 61.26 | 88.16 | 78.46 | 91.90 | 81.06 | 91.80 |
| | UnLLM | 83.90 | 93.58 | 90.00 | 92.67 | – | – |
| 0.5 | Zero-shot | 58.26 | 43.65 | 62.93 | 60.57 | 68.69 | 59.42 |
| | Few | 37.66 | 63.22 | 18.62 | 69.26 | 70.89 | 52.36 |
| | Cot | 63.62 | 50.88 | 62.88 | 48.06 | 74.93 | 71.44 |
| | Analogy | 56.35 | 76.52 | 72.87 | 84.55 | 83.53 | 87.60 |
| | UnLLM | 93.42 | 93.00 | 96.63 | 95.63 | – | – |
| 0.75 | Zero-shot | 57.24 | 31.56 | 67.00 | 50.47 | 75.11 | 51.93 |
| | Few-shot | 44.57 | 34.87 | 15.39 | 42.20 | 75.87 | 46.00 |
| | Cot | 65.62 | 40.10 | 66.75 | 37.32 | 77.44 | 62.28 |
| | Analogy | 47.60 | 50.13 | 71.27 | 64.58 | 83.68 | 73.61 |
| | UnLLM | 96.94 | 89.73 | 96.58 | 89.17 | – | – |

is generally stronger than other in-context strategies, showing clear gains, highlighting that the designed analogy-augmented inference is an effective complement to prompting. By contrast, Un-LLM achieves consistently superior K-F1 and N-F1 scores on both LLaMA3.1-8B and Qwen2.5-32B, regardless of the ratio, and significantly outperforms all prompting-based methods. We do not

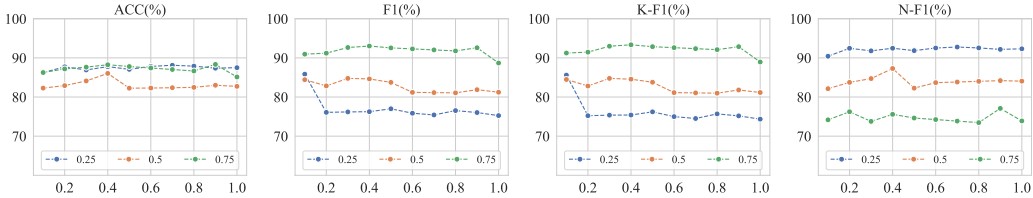

Figure 3: Parameter sensitivity analysis of $\lambda_{\text{orth}}$ on the BANKING dataset.

fine-tune DeepSeek due to its extremely high GPU requirements. For completeness, we report other evaluation metrics (ACC, F1) in the Appendix C.3.

## 5.4 QUALITATIVE ANALYSIS

**Parameter Sensitivity Analysis**  To assess the impact of $\lambda_{\text{orth}}$, we conduct experiments on the BANKING dataset across three proportions, varying $\lambda_{\text{orth}}$ from 0.1 to 1.0 (Figure 3). The results indicate that the orthogonal loss effectively improves both ID classification and OOD detection, with $\lambda_{\text{orth}} = 0.4$ yielding relatively balanced performance across various evaluation metrics and proportions. Overall, the model demonstrates robustness to this parameter selection, showing that it not only enhances OOD detection but also benefits ID classification. More sensitivity analyses on other parameters are provided in Appendix B.4.

**Ablation Study**  To evaluate the effectiveness of our proposed modules, we compared Un-LLM with several variants: "w/o $\mathcal{L}_{\text{cl}}$" that excludes $\mathcal{L}_{\text{cl}}$ during training, "w/o $\mathcal{L}_{\text{orth}}$" that removes $\mathcal{L}_{\text{orth}}$, "w/o calibration" that eliminates OOD parameter calibration, and "w/o analogy" performs infrence without analogy-augmented self reflection.  As shown in Figure 4, Un-LLM achieves the best performance across all ratios.  Below, we analyze the role and significance of each component:

First, removing the contrastive loss leads to a significant performance drop, particularly when the ratio is 0.25 or 0.5.  This result highlights the importance of contrastive learning in promoting tighter intra-class clustering, which is crucial for distinguishing ID and OOD representations.  Second, as the ratio increases, the

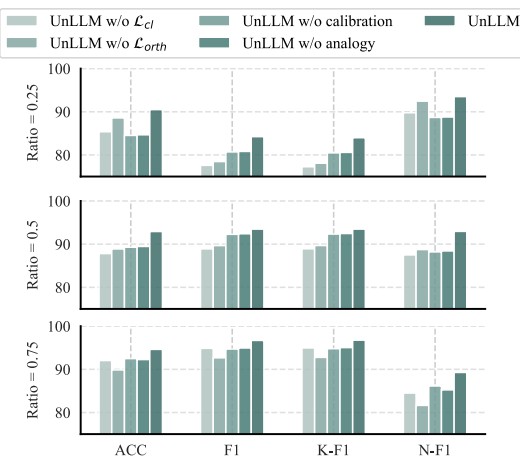

Figure 4: Ablation study on CLINC dataset.

impact of orthogonal constraint loss becomes more pronounced.  A higher ratio introduces more labels, increasing the likelihood of overfitting on known categories. The orthogonal constraint loss effectively mitigates this risk by enhancing the separation between ID and OOD representations. Finally, OOD parameter calibration and analogy-augmented self-reflection prove to be indispensable, as their removal results in a substantial performance decline. This finding underscores the importance of addressing knowledge-output misalignment and overconfidence in semantically similar labels for improving OOD detection. Due to page limitations, additional ablation study results on other datasets are provided in Appendix C.1.

## 5.5 CASE STUDY

We compare UnLLM with the LLM-OOD baseline on StackOverflow (known-class ratio 0.75). Figure 5 visualizes the learned representations via T-SNE: UnLLM forms more compact ID clusters and maintains a clearer margin to OOD samples, indicating improved representation-level separability.

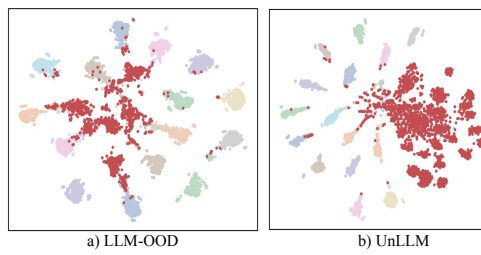

a) LLM-OOD            b) UnLLM

Figure 5: Representation visualization with red denoting OOD and other colors representing ID.

Table 3: Performance on Intra–Inter Ratio with different methods. Higher is better.

| Dataset | Ratio | LLM-OOD | UnLLM |
|---------|-------|---------|-------|
| StackOverflow | 0.25 | 0.99 | 1.98 |
| | 0.50 | 0.88 | 5.62 |
| | 0.75 | 0.68 | 4.56 |
| CLINC | 0.25 | 0.90 | 1.15 |
| | 0.50 | 0.94 | 21.30 |
| | 0.75 | 1.09 | 6.43 |

To complement this qualitative evidence, we further report the Intra–Inter Ratio, which quantifies ID–OOD separation in the representation space:

$$\text{Intra–Inter Ratio} = \frac{\text{Inter-cluster Distance}}{\text{Intra-cluster Distance}} = \frac{\text{Avg. cosine distance between ID and OOD clusters}}{\text{Avg. cosine distance within the ID cluster}}. \tag{6}$$

A higher ratio suggests that ID samples are tightly clustered while remaining far from OOD samples. As summarized in Table 3, UnLLM consistently yields higher ratios than LLM-OOD, confirming its improved ID–OOD separability across datasets and settings.

## 6 CONCLUSION

In this work, we proposed UnLLM, for OSTC, addressing key limitations stemming from the absence of supervised signals for unknown instances during training. First, we introduced an open-set generative fine-tuning to develop a K+1 classifier. Additionally, we proposed an OOD parameter calibration method to align the model's internal cognition of the unknown with its outputs. During inference, we integrated an analogy-augmented self-reflection mechanism to mitigate overconfidence. Experimental results across six datasets demonstrate that UnLLM consistently achieves SOTA performance in OOD detection while maintaining competitive accuracy in ID classification.

## 7 ETHICS STATEMENT

We acknowledge and adhere to the ICLR Code of Ethics. We commit to promoting positive societal impact while considering and mitigating potential harms (e.g., privacy, bias, unfairness). We uphold scientific integrity by ensuring transparency, reproducibility, and honest reporting of methods and limitations, and we disclose any ethical review status (e.g., IRB approval) when applicable. We further disclose conflicts of interest, respect fairness and non-discrimination, and give proper credit to prior work while honoring confidentiality and data usage rights.

## 8 REPRODUCIBILITY STATEMENT

We summarize our efforts below to facilitate reproducible results:

- **Datasets.** We use publicly available datasets, which are described in detail in Section 5.1, Section 5.1, and Appendix B.1.

- **Baselines.** The description and hyperparameters of the OOD detection baselines are explained in Section 5.1 and Appendix B.3.

- **Methodology.** Our method is fully documented in Section 4, with the pseudo algorithm detailed in Algorithm S1 and Algorithm S2. Hyperparameters are specified in Section B.4, with a thorough ablation study provided in Section 5.4 and Appendix C.1.

- **Open Source.** Code and datasets are publicly available at `https://github.com/cx9941/UnLLM`.

ACKNOWLEDGMENTS

This work was supported in part by the National Natural Science Foundation of China (NSFC) (Grant No. 62506352), in part by the Strategic Priority Research Program of Chinese Academy of Sciences (Grant No. XDB1350102), in part by the National Natural Science Foundation of China (Grant No.92370204), in part by the National Key R&D Program of China (Grant No.2023YFF0725001), in part by the guangdong Basic and Applied Basic Research Foundation (Grant No.2023B1515120057), in part by the Key-Area Special Project of Guangdong Provincial Ordinary Universities (2024ZDZX1007), in part by the Natural Science Foundation of Anhui Province (Grant No. 2508085QF211), the National Natural Science Foundation of China (Grant No. 62506348), the Opening Foundation of State Key Laboratory of Cognitive Intelligence, iFLYTEK (COGOS-2025HE02).

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

# A  MORE DETAILS OF UNLLM

## A.1  LABEL NUMBERING METHODOLOGY

To standardize label formatting, we employ a zero-padded numbering scheme with three digits. Specifically, the numerical identifier $\mathrm{NID}(x)$ for a label $x$ is computed as: $\mathrm{NID}(x) = \mathrm{str.zfill}(x, 3)$, where $\mathrm{str.zfill}(x, i)$ converts the label $x$ into a zero-padded string of length $i$. In this work, we set $i = 3$. For example, given the label set $\{1, 2, 3\}$, the corresponding identifiers are:

$$
\begin{aligned}
\mathrm{ID}(1) &= \mathrm{str.zfill}(1, 3) = \text{``001''}, \\
\mathrm{ID}(2) &= \mathrm{str.zfill}(2, 3) = \text{``002''}, \\
\mathrm{ID}(3) &= \mathrm{str.zfill}(3, 3) = \text{``003''}.
\end{aligned}
\tag{7}
$$

This numbering scheme ensures a consistent label representation, facilitating structured data processing and representation extraction.

## A.2  PRINCIPAL COMPONENT ANALYSIS FOR OOD SUBSPACE CONSTRUCTION

We formalize the construction of the OOD subspace using PCA. Let the virtual outliers sampled across all classes form the set $\mathcal{V}$, whose flattened feature embeddings are stacked into the matrix $\mathbf{M} \in \mathbb{R}^{|\mathcal{V}| \times td}$. The covariance matrix is computed as $\mathbf{\Sigma}_{\mathrm{neg}} = \mathbf{M}^\top \mathbf{M} \in \mathbb{R}^{td \times td}$. Applying singular value decomposition (SVD), we obtain $\mathbf{\Sigma}_{\mathrm{neg}} = \mathbf{U}\mathbf{\Lambda}\mathbf{U}^\top$, where columns of $\mathbf{U}$ are orthogonal eigenvectors representing the dominant directions of variation in $\mathcal{V}$. Selecting the top $e$ eigenvectors yields the OOD subspace $\mathbf{O} = \mathbf{U}[:, :e] \in \mathbb{R}^{td \times e}$. ID features $\mathbf{H}_{\mathrm{ID}} \in \mathbb{R}^{N_{\mathrm{ID}} \times td}$ are projected onto $\mathbf{O}$, and the orthogonality loss of the main text penalizes their alignment, thereby promoting a clearer separation between ID and OOD spaces.

## A.3  EXAMPLES FOR LLM INPUTS

To better help readers understand the prompt, we give two specific instances for LLM training and inference with self-reflection as follows:

---

**Training Instance**

**You are an expert in text classification with a focus on Out-of-Distribution (OOD) detection.**

**Your task is to accurately classify a given piece of text into one of the provided categories only if it strictly matches the meaning and scope of the category definition. If the text does not match any category definition, it should be identified as "Out-of-Distribution (OOD)"**

**Text to Classify:**
*Typed FP: Tuple Arguments and Curriable Arguments.*

**Questions: Given the candidate categories:** *["000.It is OOD", "002.haskell", "003.magento", "005.apache"],* **which category does the text belong to?**

**Answer:** *002.haskell*.

---

---

**Self-Reflection Inference Instance**

**You are an expert in text classification with a focus on Out-of-Distribution (OOD) detection.**

**Your task is to accurately classify a given piece of text into one of the provided categories only if it strictly matches the meaning and scope of the category definition. If the text does not match any category definition, it should be identified as "Out-of-Distribution (OOD)"**

**Text to Classify:**
*Think someone has took money out with my card. What shall I do?*

**Questions: Given the candidate categories:** *["012.top up reverted", "010.declined card payment", "019.compromised card", "018.cash withdrawal charge", "011.reverted card payment?", "000.It is OOD", "007.failed transfer", "008.edit personal details", "005.card arrival", "016.card not working", "002.balance not updated after cheque or cash deposit"],* **which category does the text belong to?**

**Answer:** *019.compromised card.*.

**Recall relevant exemplars of** *019.compromised card.*: *["I sees some suspicious spending on my credit card that I don't recall I had made. What should I do?", "Someone might be using my card. What should I do?", ...]*
**Does the text strictly align with the specified scope of** *019.compromised card.*? **Please start by answering Yes or No.**
**Answer:** *No*.

---

## A.4 ALGORITHM PSEUDOCODE

In this paper, we designed a novel method to implement the OSTC. In order to make our algorithm procedure easier to understand, the pseudocode for the fine-tuning pipeline of open-set generative fine-tuning is available in Algorithm S1 and the inference pipeline with analogy-augmented self-reflection is provided in Algorithm S2.

---

**Algorithm S1** LLM Fine-Tuning Procedure

---

**Require:** Pre-trained LLM, Training set $\{(x_i, y_i)\}_{i=1}^N$, Number of partitions $s$, Hyperparameters $\lambda_{\text{cl}}, \lambda_{\text{orth}}$
1: Initialize model parameters $\theta$
2: **for** each batch $\{(x_i, y_i)\}_{i \in B}$ **do**
3:     **for** each training instance $x_i$ in the batch **do**
4:         Partition label set $\mathcal{Y}$ into $s$ mutually exclusive subsets $\{\mathcal{Y}_{i,j}^p\}_{j=1}^s$
5:         **for** each label partition $\mathcal{Y}_{i,j}^p$ **do**
6:             Generate partition-conditional label $\tilde{y}_{i,j}$
7:             Compute generative loss $\mathcal{L}_{\text{gen}}$ (Eq. 1)
8:             Extract representation $\mathbf{h}_{i,j}$
9:         **end for**
10:     **end for**
11:     Compute contrastive loss $\mathcal{L}_{\text{cl}}$ (Eq. 2)
12:     Estimate class-conditional distribution parameters $\widehat{\mu}_k, \widehat{\sigma}_k^2$
13:     Sample virtual outliers $\mathbf{v}_k$ from low-likelihood regions
14:     Compute orthogonality constraint loss $\mathcal{L}_{\text{orth}}$
15: **end for**
16: Optimize LLM parameters $\theta$ using composite loss $\mathcal{L}$
17: **return** Fine-tuned LLM

---

---

**Algorithm S2** LLM Inference Procedure

---

**Require:** Fine-tuned LLM, Test instance $x_i$
  1: Partition label set $\mathcal{Y}$ into $s$ mutually exclusive subsets $\{\mathcal{Y}^p_{i,j}\}^s_{j=1}$
  2: **for** each label partition $\mathcal{Y}^p_{i,j}$ **do**
  3:    Construct textual input with candidate labels
  4:    Generate predicted label $\hat{y}_{i,j}$
  5:    **if** $\hat{y}_{i,j} \neq K + 1$ **then**
  6:      **return** $\hat{y}_{i,j}$ (ID prediction)
  7:    **end if**
  8: **end for**
  9: Retrieve analogical examples $\{a_1, a_2, \dots\}$ based on similarity
 10: Perform analogy-augmented self-reflection
 11: **If** response is "No", classify as $K + 1$
 12: **return** $\hat{y}_{i,j}$ (ID prediction)

---

Table S1: Statistics of datasets. $\|\|$ denotes the total number of instances. Length indicates the average length of each instance in the dataset.

| Dataset | N | $\|$Train$\|$ | $\|$Val$\|$ | $\|$Test$\|$ | Length |
|---|---|---|---|---|---|
| BANKING | 77 | 9,003 | 1,000 | 3,078 | 11.77 |
| CLINC | 150 | 17,995 | 2,250 | 2,250 | 8.31 |
| StackOverflow | 20 | 11,996 | 1,998 | 5,991 | 8.34 |
| Reviews | 50 | 29,823 | 4,942 | 14,794 | 143.04 |
| Newsgroups | 20 | 11,291 | 1880 | 5,657 | 306.75 |
| THUCNews | 14 | 25,200 | 2,800 | 5,600 | 7.15 |

Table S2: Dataset Splits and Statistics. $\|\|$ denotes the total number of instances.

| Ratio | 0.25 | | | 0.50 | | | 0.75 | | |
|---|---|---|---|---|---|---|---|---|---|
| Dataset | $\|$Train$\|$ | $\|$Val$\|$ | $\|$Test$\|$ | $\|$Train$\|$ | $\|$Val$\|$ | $\|$Test$\|$ | $\|$Train$\|$ | $\|$Val$\|$ | $\|$Test$\|$ |
| BANKING | 2,222 | 247 | 3,078 | 4,592 | 511 | 3,078 | 6,784 | 755 | 3,078 |
| CLINC | 4,439 | 555 | 2,250 | 8,996 | 1,125 | 2,250 | 13,436 | 1,680 | 2,250 |
| StackOverflow | 2,998 | 499 | 5,991 | 5,999 | 999 | 5,991 | 8,998 | 1,499 | 5,991 |
| Reviews | 7,176 | 1,192 | 14,794 | 14,935 | 2,465 | 14,794 | 22,649 | 3,750 | 14,794 |
| Newsgroups | 2,835 | 472 | 5,657 | 5,876 | 979 | 5,657 | 8,578 | 1,427 | 5,657 |
| THUCNews | 5,400 | 600 | 5,600 | 12,600 | 1,400 | 5,600 | 19,800 | 2,200 | 5,600 |

# B   MORE DETAILS ON EXPERIMENTAL SETTINGS

## B.1   DATASET DESCRIPTION

To verify the effectiveness and universality of our proposed method, we conducted exhaustive experiments on six widely used text classification datasets. These datasets are summarized as:

- **Newsgroups** (Schneider, 2003) consists of 18,828 documents partitioned evenly across 20 mutually exclusive classes.

- **Reviews** (Jindal & Liu, 2008) consists of 50 classes of products or domains, each with 1,000 review documents.

- **CLINC** (Larson et al., 2019) is a very popular dataset, which encompasses a broad range of intents, totaling 150 across 10 domains. The entire dataset consists of 22500 in-domain samples and 1200 Out-of-domain samples.

- **BANKING** (Casanueva et al., 2020) is a kind of dataset about the banking business, with 77 categories. The data is characterized by the imbalance of samples in different categories. The training set, validation set, and test set contain 9003, 1000, and 3080 samples respectively.

- **StackOverflow** (Xu et al., 2015) is a dataset about programming languages released by Kaggle.com. The dataset is subdivided into 20 categories and has 2 samples. The number of samples in the training set, validation set, and test set is 12000, 2000, and 6000 respectively.
- **THUCNews** (Li et al., 2006) is a Chinese news text dataset, with 10 categories: finance, real estate, stocks, education, technology, society, politics, sports, games, and entertainment.

These datasets are only for scientific research and are available for all members of the NLP research community. We have adhered to the typical method of utilizing these resources.

### B.2 MORE DETAILS ON DATASET SPLIT

We present the dataset split statistics across different training ratios in Table S2. The training and validation sets contain only ID samples, while the test set includes both ID and OOD samples.

### B.3 MORE DETAILS ON BASELINES

We categorize the baselines into three groups: *backbone-specific classification models*, *OOD detection methods*, and *confidence calibration methods*. The former rely on model-specific training objectives, while the latter two are model-agnostic techniques that can be applied on top of different backbones (CNN, LSTM, PLMs, and LLMs).

**Backbone-specific methods.**

- **DOC** (Shu et al., 2017): A CNN-based classifier replacing softmax with one-vs-rest sigmoid outputs to reduce open-space risk.
- **DeepUnk** (Lin & Xu, 2019): An LSTM-based model that learns deep intent features using large-margin cosine loss.
- **ADB** (Zhang et al., 2021): A BERT-based method that learns adaptive spherical decision boundaries for open intent detection.
- **CLAP** (Liu et al., 2023): A BERT-based method that inflates and shrinks decision boundaries to balance ID/OOD separation.
- **KNNCon** (Zhou et al., 2022): A BERT-based method that leverages KNN-based contrastive learning during training and applies LOF for OOD scoring.
- **DyEn** (Zhou et al., 2023): A BERT-based method that dynamically ensembles internal classifiers and uses early exits to mitigate PLM overthinking.
- **LLM-OOD** (Liu et al., 2024): An LLM-based method that reformulates classification as text generation, leveraging LLM fine-tuning to produce more isotropic embeddings for OOD detection.

**OOD detection methods (model-agnostic).** These methods provide scoring mechanisms that can be applied on top of different backbones:

- **OpenMax** (Bendale & Boult, 2016): Fits Weibull distributions to logits to estimate open-set probabilities.
- **Energy** (Liu et al., 2020): Uses energy-based confidence scoring to mitigate overconfidence.
- **VOS** (Du et al., 2022): Generates virtual Gaussian-based outliers for decision boundary regularization.
- **NPO** (Tao et al., 2023): Improves on VOS by synthesizing non-parametric outliers without distributional assumptions.
- **ViM** (Wang et al., 2022): Computes virtual logit margins to enhance ID/OOD separation.
- **SHE** (Burns & Fukai, 2023): Extends Hopfield networks with higher-order (setwise) connections encoded via a simplicial complex, increasing memory storage capacity and strengthening attractor dynamics; also instantiated with continuous Hopfield networks, suggesting potential improvements to Transformer attention.

**Confidence calibration methods (model-agnostic).** These methods aim to improve the reliability of predictive probabilities and can be applied on top of different backbones:

- **Temperature Scaling** (Guo et al., 2017): A simple yet effective post-hoc calibration technique that rescales logits with a single temperature parameter to alleviate overconfidence.

- **LogitNorm** (Wei et al., 2022): A training-time regularization method that normalizes logits across samples, encouraging more consistent confidence estimates and improving model calibration.

**Implementation details.** All *backbone-specific methods* are reproduced strictly following the parameter settings reported in their original papers. For *OOD detection methods* and *confidence calibration methods*, we implement them within a unified framework to ensure consistency across backbones. It is worth noting that most of these methods were originally developed in the computer vision community and have not been directly applied to open-set text classification (OSTC). Therefore, we adapt their formulations to the OSTC setting and re-implement them based on the pytorch_ood[1] library for a fair comparison. For these these methods, we first fine-tuned BERT or LLM on the training set and classify samples with an OOD score exceeding a predefined threshold as OOD. Following prior work (Zeng et al., 2021), we set this threshold to 0.5.

### B.4 MORE IMPLEMENT DETAILS

**Implement Details** For a fair comparison, we used bert-base-uncased[2] and bert-base-chinese[3] as backbones for discriminative language models on English and Chinese datasets, respectively, and LLaMA3.1-8B-Instruct[4] as the backbone for generative models. For BERT (Devlin, 2018), we applied a full-parameter fine-tuning method for training. For LLaMA (Touvron et al., 2023), we employed QLoRA (Dettmers et al., 2024) to minimize additional parameter requirements during training. We adopted AdamW optimizer (Loshchilov & Hutter, 2019) and set learning rate as 1e-4, the training batch size as 16, the epochs 3, label partition number $s$ as 2, and virtual OOD subspace dimension $e$ as 20. All experiments were conducted on a machine equipped with 8 NVIDIA A100 GPUs (each with 40GB memory). The training was performed for 3 epochs for LLM-based methods and 20 epochs for non-LLM-based methods. Unless otherwise stated, training is conducted with a learning rate of $5 \times 10^{-5}$, a batch size of 16 for training, and a batch size of 32 for evaluation. For LLM training, we utilized a parameter-efficient fine-tuning (PEFT) method to minimize additional parameter requirements. Specifically, we employed the QLoRA technique (Dettmers et al., 2024), which freezes the weights of pre-trained LLMs and integrates trainable low-rank decomposition matrices into each Transformer layer. During fine-tuning, the model predicted answers using only the class label tokens to compute the auto-regressive loss. During inference, we utilized the BERT pooler layer as the PLM embedding function in Eq 5.

## C MORE DETAILS ON EXPERIMENTAL RESULTS

### C.1 MORE DETAILS FOR ABLATION STUDIES

In addition to the ablation study on the CLINC dataset, we also conduct the same experiments on the StackOverflow and BANKING datasets. The results, summarized in Figure S1a and Figure S1b, are consistent with our previous observations, further validating the effectiveness of the modules.

We provide additional ablation evidence to further verify that our reformulation (Open-set SFT) is essential and serves as the foundation for subsequent modules. To avoid ambiguity, we distinguish two training paradigms:

- **Closed-set SFT** fine-tunes the LLM only on ID labels, optimizing it to generate ID class outputs exclusively; no OOD semantics are introduced during training.

---

[1]https://github.com/kkirchheim/pytorch-ood

[2]https://huggingface.co/google-bert/bert-base-uncased

[3]https://huggingface.co/google-bert/bert-base-chinese

[4]https://huggingface.co/meta-llama/Llama-3.1-8B-Instruct

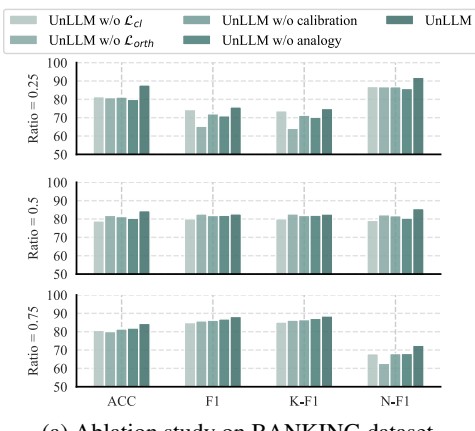
(a) Ablation study on BANKING dataset.

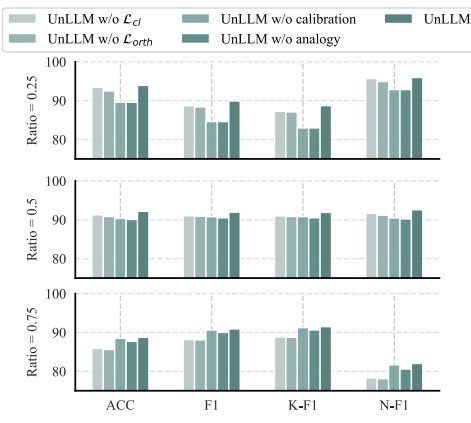
(b) Ablation study on StackOverflow dataset.

Figure S1: Ablation study on BANKING and StackOverflow datasets.

Table S3: Extended ablation study on CLINC with known-class ratio 0.25.

| Category | Method | ACC(%) | F1(%) | K-F1(%) | N-F1(%) |
|---|---|---|---|---|---|
| LLM Output | Closed-set SFT | 24.31 | 48.40 | 49.71 | 0.00 |
| | + Contrastive Learning | 24.53 | 48.50 | 49.80 | 0.35 |
| | + Contrastive Learning + Orthogonal Regularizer | 24.58 | 47.53 | 48.79 | 0.82 |
| | + Contrastive Learning + Orthogonal Regularizer + Analogy | 36.89 | 43.81 | 43.98 | 37.27 |
| Post-hoc | LLM-OOD | 65.66 | 66.68 | 66.65 | 67.89 |
| | + Contrastive Learning | 68.00 | 30.21 | 29.71 | 78.26 |
| | + Contrastive Learning + Orthogonal Regularizer | 81.02 | 74.33 | 73.68 | 86.28 |
| | + Contrastive Learning + Orthogonal Regularizer + Analogy | 80.80 | 73.49 | 73.15 | 86.16 |
| LLM Output | Open-set SFT | 80.18 | 75.50 | 75.24 | 85.30 |
| | + Contrastive Learning | 82.22 | 77.57 | 77.32 | 86.88 |
| | + Contrastive Learning + Orthogonal Regularizer | 83.51 | 78.98 | 78.73 | 88.09 |
| | + Contrastive Learning + Orthogonal Regularizer + Calibration | 85.09 | 80.32 | 80.09 | 90.04 |
| | + Contrastive Learning + Orthogonal Regularizer + Calibration + Analogy | 89.87 | 83.28 | 83.01 | 93.15 |

- **Open-set SFT** fine-tunes the LLM with both ID labels and conditional-OOD labels, enabling the model to explicitly learn an additional OOD class during training.

**Setup.** Starting from either Closed-set SFT or Open-set SFT, we sequentially integrate the proposed modules (contrastive learning, orthogonal regularization, calibration, and analogy-augmented self-reflection) and compare against the post-hoc baseline (LLM-OOD). Table S3 reports results on CLINC with known-class ratio 0.25.

**Results and discussion.** Table S3 shows that stacking modules on top of *Closed-set SFT* does not reliably alleviate the large-scale overfitting of LLM outputs to the ID label space; performance can even degrade (e.g., F1 from 48.40 to 43.81). In contrast, *Open-set SFT* alone yields a substantial gain (F1 from 48.40 to 75.50), and the subsequent modules further improve performance, reaching 83.28 F1. These results support our claim that Open-set SFT provides the crucial mechanism that makes the later components effective.

## C.2 MORE DETAILS FOR PARAMETER SENSITIVITY ANALYSIS

Besides $\lambda_{\mathrm{orth}}$, we also conducted parameter sensitive analysis of $\lambda_{\mathrm{cl}}$, and $|\mathcal{V}_k|$.

**The Influence of $\lambda_{\mathbf{cl}}$** We analyzed the impact of $\lambda_{\mathrm{cl}}$ on model performance by varying it from 0.1 to 1.0 (Figure S2a). Results show that contrastive loss is effective for both ID classification and OOD detection, with the balanced performance at $\lambda_{\mathrm{cl}} = 0.8$, enhancing ID-OOD separation. However, higher $\lambda_{\mathrm{cl}}$ values harm performance, as orthogonal loss compresses ID representations, weakening intra-ID classification. This highlights the necessity of balancing generative and contrastive objectives for optimal results.

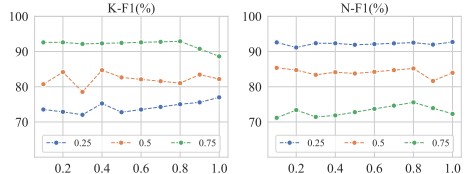
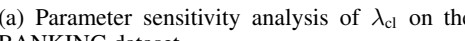
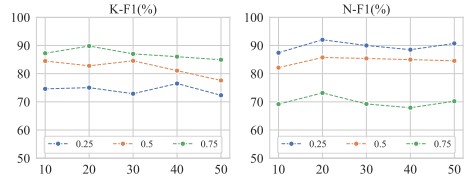

(a) Parameter sensitivity analysis of $\lambda_{cl}$ on the BANKING dataset.

(b) Parameter sensitivity analysis of $|\mathcal{V}_k|$ on the BANKING dataset.

Figure S2: Ablation study on different datasets.

Table S4: Performance comparison on CLINC dataset across different prompting strategies.

| Backbone | Method | 0.25 | | | | 0.5 | | | | 0.75 | | | |
|---|---|---|---|---|---|---|---|---|---|---|---|---|---|
| | | ACC | F1 | K-F1 | N-F1 | ACC | F1 | K-F1 | N-F1 | ACC | F1 | K-F1 | N-F1 |
| LLaMA3.1-8B | Zero-Shot | 53.05 | 52.04 | 51.85 | 59.37 | 50.42 | 58.07 | 58.26 | 43.65 | 54.38 | 57.01 | 57.24 | 31.56 |
| | Few-shot | 55.40 | 49.81 | 49.45 | 63.01 | 54.24 | 38.00 | 37.66 | 63.22 | 45.18 | 44.48 | 44.57 | 34.87 |
| | Cot | 50.60 | 54.99 | 55.00 | 54.43 | 57.09 | 63.45 | 63.62 | 50.88 | 62.64 | 65.40 | 65.62 | 40.10 |
| | Analogy | 61.97 | 81.96 | 61.26 | 88.16 | 56.61 | 69.78 | 56.35 | 76.52 | 47.62 | 52.04 | 47.60 | 50.13 |
| Qwen2.5-32B | Zero-Shot | 56.11 | 57.77 | 57.65 | 62.20 | 61.17 | 62.90 | 62.93 | 60.57 | 65.26 | 66.85 | 67.00 | 50.47 |
| | Few-shot | 74.32 | 19.27 | 17.52 | 84.30 | 55.40 | 19.29 | 18.62 | 69.26 | 34.25 | 15.63 | 15.39 | 42.20 |
| | Cot | 52.42 | 56.84 | 56.86 | 56.30 | 56.86 | 62.68 | 62.88 | 48.06 | 62.91 | 66.49 | 66.75 | 37.32 |
| | Analogy | 78.81 | 88.22 | 78.46 | 91.90 | 73.02 | 81.38 | 72.87 | 84.55 | 71.21 | 72.09 | 71.27 | 64.58 |
| DeepSeek-V3-0324 | Zero-Shot | 57.11 | 60.33 | 58.70 | 61.90 | 64.53 | 69.47 | 68.69 | 59.42 | 71.42 | 75.57 | 75.11 | 51.93 |
| | Few-shot | 54.71 | 60.33 | 58.81 | 57.87 | 62.74 | 71.58 | 70.89 | 52.36 | 71.56 | 76.27 | 75.87 | 46.00 |
| | Cot | 60.71 | 62.75 | 61.03 | 65.54 | 73.24 | 75.87 | 74.93 | 71.44 | 75.91 | 77.99 | 77.44 | 62.28 |
| | Analogy | 81.35 | 88.10 | 81.06 | 91.80 | 83.59 | 86.01 | 83.53 | 87.60 | 83.59 | 82.05 | 83.68 | 73.61 |
| LLaMa3.1-8B | UnLLM | 90.58 | 84.16 | 83.90 | 93.58 | 92.96 | 93.42 | 93.42 | 93.00 | 94.87 | 96.88 | 96.94 | 89.73 |
| Qwen2.5-7B | UnLLM | 87.82 | 80.59 | 80.29 | 91.88 | 88.80 | 93.88 | 93.96 | 87.76 | 94.44 | 96.37 | 96.42 | 90.54 |
| Qwen2.5-32B | UnLLM | 89.42 | 90.07 | 90.00 | 92.67 | 95.51 | 96.62 | 96.63 | 95.63 | 94.13 | 96.51 | 96.58 | 89.17 |

**The Influence of $|\mathcal{V}_k|$** We examined the impact of the number of virtual outliers per class $|\mathcal{V}_k|$ in Figure S2b, where we vary $|\mathcal{V}_k|$ across $\{10, 20, 30, 40, 50\}$. In general, a larger $|\mathcal{V}_k|$ is advantageous as it allows for a more accurate estimation of Gaussian distribution parameters. However, in many cases, the model struggles to sample a sufficient number of negative instances within the designated low-confidence region, leading to a decline in performance. Consequently, we set the queue size to 20 in our experiments. Nevertheless, in some cases, a smaller queue size is necessary due to the limited number of instances available for certain classes.

## C.3 MORE DETAILS OF RESULTS ON UNLLM VS. LLMS WITH DIFFERENT PROMPTING STRATEGIES

Table S4 reports a detailed comparison between UnLLM and strong prompting baselines on CLINC under different known-class ratios. We consider standard prompting strategies, including zero-shot, few-shot, and chain-of-thought (CoT), as well as the analogy prompting used in our inference-time self-reflection. Overall, prompting alone is insufficient to provide robust open-set performance: while analogy prompting can improve results for some backbones and ratios, its behavior is unstable across settings and may degrade as the known-class ratio increases. In contrast, UnLLM achieves consistently strong performance across backbones and ratios, demonstrating that the gains mainly come from our training-time reformulation and learning objectives rather than prompt engineering.

## C.4 RESULTS ON UNLLM VS. HALLUCINATION DETECTION METHODS

Another line of research that appears related to OSTC is hallucination detection in LLMs. The relevance lies in their shared focus on uncertainty estimation—both tasks aim to identify model outputs that deviate from expected or reliable content. However, a fundamental distinction exists: hallucination detection primarily targets generation-based inconsistencies, often evaluating factual correctness in open-ended text generation, whereas OSTC is a classification task that jointly models in-distribution classification and OOD detection. Accurate class prediction is a prerequisite for

Table S5: Performance Comparison vs. hallucination detection methods

| Ratio | Method | BANKING | | | | CLINC | | | | StackOverflow | | | |
|---|---|---|---|---|---|---|---|---|---|---|---|---|---|
| | | ACC | F1 | K-F1 | N-F1 | ACC | F1 | K-F1 | N-F1 | ACC | F1 | K-F1 | N-F1 |
| 0.25 | LLM-Check (Attn Score) | 71.09 | 15.54 | 14.33 | 82.74 | 61.8 | 26.4 | 36.45 | 74.56 | 74.98 | 14.61 | 0.33 | 85.69 |
| | LLM-Check (Hidden Score) | 69.82 | 16.55 | 15.78 | 81.8 | 69.46 | 15.41 | 16.66 | 81.45 | 75.00 | 14.72 | 0.44 | 85.7 |
| | CED Score | 59.23 | 5.04 | 1.38 | 74.45 | 58.31 | 2.34 | 0.41 | 73.64 | 67.5 | 13.64 | 0.25 | 80.59 |
| | HaluEval (COT) | 50.63 | 49.81 | 52.09 | 56.32 | 50.60 | 54.99 | 55.00 | 54.43 | 52.74 | 55.41 | 55.39 | 55.53 |
| | UnLLM | 87.86 | 75.89 | 75.04 | 92.02 | 90.58 | 84.16 | 83.90 | 93.58 | 94.00 | 89.87 | 88.65 | 96.00 |
| 0.5 | LLM-Check (Attn Score) | 50.16 | 16.78 | 16.6 | 64.68 | 50.89 | 19.84 | 20.70 | 64.57 | 49.91 | 6.17 | 0.12 | 66.57 |
| | LLM-Check (Hidden Score) | 51.04 | 13.07 | 11.98 | 66.17 | 48.76 | 15.77 | 16.33 | 63.15 | 49.91 | 6.17 | 0.12 | 66.57 |
| | CED Score | 45.61 | 2.07 | 0.47 | 62.64 | 39.16 | 1.02 | 0.29 | 56.24 | 44.52 | 5.65 | 0.05 | 61.63 |
| | HaluEval (COT) | 51.58 | 51.79 | 53.21 | 49.53 | 57.09 | 63.45 | 63.62 | 50.88 | 69.06 | 75.02 | 76.45 | 60.75 |
| | UnLLM | 84.57 | 82.82 | 82.74 | 85.72 | 92.96 | 93.42 | 93.42 | 93.00 | 92.20 | 91.97 | 91.91 | 92.60 |
| 0.75 | LLM-Check (Attn Score) | 31.84 | 21.25 | 21.37 | 40.01 | 29.6 | 14.67 | 14.58 | 39.94 | 25.11 | 2.75 | 0.25 | 40.05 |
| | LLM-Check (Hidden Score) | 28.40 | 10.94 | 10.50 | 40.34 | 31.20 | 22.14 | 22.93 | 37.53 | 25.09 | 2.75 | 0.25 | 40.02 |
| | CED Score | 22.32 | 1.67 | 1.07 | 36.23 | 24.04 | 0.34 | 0.00 | 38.77 | 18.69 | 2.01 | 0.04 | 31.48 |
| | HaluEval (COT) | 56.87 | 61.78 | 63.4 | 31.56 | 62.64 | 65.40 | 65.62 | 40.10 | 80.52 | 84.27 | 85.73 | 62.42 |
| | UnLLM | 84.54 | 88.31 | 88.58 | 72.60 | 94.87 | 96.88 | 96.94 | 89.73 | 89.68 | 91.78 | 92.36 | 83.08 |

Table S6: Performance of EnergyBased and UnLLM under different similarity thresholds on CLINC (KCR=0.25).

| Threshold | Method | ACC(%) | F1(%) | K-F1(%) | N-F1(%) |
|---|---|---|---|---|---|
| 0.96 | EnergyBased | 84.47 | 67.78 | 66.63 | 89.45 |
| | UnLLM | 91.93 | 80.85 | 80.14 | 94.42 |
| 0.97 | EnergyBased | 76.92 | 53.65 | 49.21 | 84.75 |
| | UnLLM | 97.44 | 94.19 | 93.33 | 98.46 |

OSTC, which necessitates a different kind of uncertainty modeling that is sensitive to categorical decision boundaries rather than semantic fidelity in free-form generation.

Despite these differences, we include hallucination detection baselines in our evaluation for completeness. Specifically, we implemented representative scoring functions from LLM-Check (Sriramanan et al., 2024) (e.g., Attention Score, Hidden State Score), CED score (Lee et al., 2024), and HaluEval (Li et al., 2023) (e.g., Chain-of-Thought Consistency) on our fine-tuned LLM and evaluated them under the OSTC setting. Experimental results in Table S5 confirm that these methods are inadequate in OSTC scenarios: although some baselines can achieve up to 80% of our method's performance on N-F1, their K-F1 scores remain significantly lower. This gap highlights the difficulty of balancing classification accuracy and OOD detection, which is the unique objective and core challenge of OSTC. Moreover, these hallucination baselines fail to consider knowledge injection and decision boundary formation during fine-tuning, both of which are crucial for OOD detection in classification tasks.

## C.5 MORE DETAILS FOR CASE STUDY

We clarify that both *overconfidence* and *representation–output misalignment* are well-established phenomena in LLMs, and we further verify them quantitatively in our OSTC setting.

**Evidence for overconfidence.** LLMs often make overly confident predictions when a test utterance is semantically close to some known labels. To substantiate this, for each test sample we compute its similarity to all candidate labels and use the similarity associated with the predicted label as a *confusability* score. We then apply different similarity thresholds and retain subsets of test samples with increasingly high confusability (i.e., harder cases where energy-based post-hoc rejection is expected to be less reliable). Table S6 shows that as the threshold increases (harder subsets), the EnergyBased baseline degrades substantially, while UnLLM remains strong, directly evidencing both the existence of overconfidence and the effectiveness of our framework in mitigating it.

**Evidence for representation–output misalignment.** Prior work has shown that LLMs may internally encode correct distinctions while the output layer fails to express them (Li et al., 2024). To validate this in our setting, we compare model variants by measuring (i) **AUC**, which reflects ID–OOD separability in representation space, and (ii) **K-F1/N-F1**, which reflect output correctness on

Table S7: Calibration ablation on CLINC: AUC (representation separability) vs. K-F1/N-F1 (output correctness).

| Model Variant | 0.25 | | | 0.5 | | | 0.75 | | |
|---|---|---|---|---|---|---|---|---|---|
| | AUC(%) | K-F1(%) | N-F1(%) | AUC(%) | K-F1(%) | N-F1(%) | AUC(%) | K-F1(%) | N-F1(%) |
| Closed-set SFT | 90.79 | 49.71 | 0.00 | 92.23 | 71.96 | 0.18 | 91.63 | 87.76 | 0.00 |
| Closed-set SFT (LLM-OOD) | 90.79 | 66.65 | 67.89 | 92.23 | 78.85 | 46.07 | 91.63 | 91.96 | 60.82 |
| Open-set SFT (no calibration) | 94.12 | 77.32 | 86.88 | 97.29 | 91.29 | 86.61 | 98.70 | 94.37 | 85.12 |
| Open-set SFT (only calibration) | 96.33 | 79.30 | 88.61 | 97.47 | 91.25 | 87.21 | 98.75 | 96.62 | 88.50 |
| UnLLM | 97.29 | 83.90 | 93.58 | 98.42 | 93.42 | 93.00 | 99.13 | 96.94 | 89.73 |

ID and OOD, respectively. Results in Table S7 reveal a clear mismatch under the closed-set training paradigm: Closed-set SFT yields high AUC yet nearly zero N-F1, indicating that the model can separate ID/OOD internally but cannot express "OOD" in its outputs. Open-set SFT substantially improves both separability and output expressiveness, and calibration further tightens this alignment, consistently increasing N-F1 across ratios. Finally, UnLLM achieves the best overall alignment and performance.

### C.6 MORE DETAILS FOR OVERALL PERFORMANCE

Beyond the main text, we additionally compare a suite of *BERT-based OOD detection* baselines and *confidence calibration* methods (Temperature Scaling, LogitNorm) while temperature scaling and logit normalization are effective for probability calibration.

As shown in Tables S8-S9, model-agnostic OOD scoring helps BERT in open-set metrics but still trails end-to-end OOD-aware training, while calibration methods (Temperature Scaling, LogitNorm) primarily improve reliability of confidences with limited impact on OSTC accuracy/F1. This reinforces the need for an OOD-aware pipeline such as UnLLMfor robust OSTC.

## D LIMITATIONS

Our proposed approach achieves substantial performance gains over SOTA baselines across multiple public OSTC benchmarks, even under constrained computational resources. Due to limited access to large-scale hardware, our experiments are conducted using the parameter-efficient QLoRA fine-tuning strategy. Remarkably, this lightweight setting already leads to significant improvements over existing methods, demonstrating the strong effectiveness and practicality of our approach. However, we have not yet evaluated it under full-parameter fine-tuning. Exploring such directions constitutes an important part of our future work.

## E THE USE OF LARGE LANGUAGE MODELS STATEMENT

The authors use Large Language Models (LLMs) as an assistive tool in the preparation of this manuscript, in accordance with the ICLR 2026 policy. We use LLMs to proofread, check grammar, and refine the language in the manuscript for improved clarity and readability.

Table S8: Performance of various methods across 6 datasets at different ratios. Metrics include ACC and F1. The best results are highlighted in bold, while the second-best results are underscored. Each result represents the mean value of four repetitive experiments.

| Ratio | Backbone | Method | BANKING ACC | F1 | CLINC ACC | F1 | StackOverflow ACC | F1 | Reviews ACC | F1 | Newsgroups ACC | F1 | THUCnews ACC | F1 |
|---|---|---|---|---|---|---|---|---|---|---|---|---|---|---|
| 0.25 | CNN | DOC | 74.70 | 67.44 | 80.23 | 74.82 | 77.06 | 68.42 | 71.31 | 59.03 | _85.26_ | _68.65_ | 73.57 | 56.29 |
| | LSTM | DeepUnk | 82.20 | 71.71 | 87.36 | 75.74 | 83.22 | 70.75 | 47.92 | 45.60 | 64.86 | 58.91 | 43.86 | 42.73 |
| | BERT | OpenMax | 59.02 | 33.72 | 64.28 | 33.89 | 32.49 | 43.30 | 31.52 | 38.26 | 32.79 | 40.22 | 49.33 | 48.16 |
| | | Energy | 60.16 | 57.43 | 74.04 | 71.63 | 60.63 | 62.30 | 38.82 | 42.20 | 61.71 | 58.97 | 39.67 | 45.92 |
| | | VOS | 38.76 | 37.03 | 51.14 | 54.04 | 75.58 | 67.88 | 73.16 | 51.57 | 49.71 | 49.74 | 68.12 | 62.12 |
| | | ViM | 52.66 | 52.26 | 60.73 | 61.24 | 54.36 | 56.72 | 57.95 | 50.19 | 42.53 | 46.78 | 75.46 | 68.55 |
| | | SHE | 43.03 | 52.45 | 48.68 | 55.93 | 36.38 | 45.59 | 39.33 | 43.33 | 39.81 | 46.21 | 51.02 | 48.89 |
| | | TemperatureScaling | 50.80 | 37.45 | 53.50 | 53.62 | 40.68 | 48.92 | 48.88 | 43.08 | 34.70 | 40.26 | 37.72 | 39.99 |
| | | LogitNorm | 70.35 | 21.71 | 74.30 | 45.92 | 51.44 | 55.78 | 57.92 | 39.50 | 45.60 | 44.40 | 22.32 | 28.31 |
| | | NPO | 42.87 | 37.31 | 56.00 | 56.45 | 76.44 | 67.03 | 74.98 | 49.07 | 51.82 | 50.80 | 68.54 | 62.99 |
| | | ADB | 54.58 | 51.61 | 60.22 | 59.20 | 83.12 | 77.99 | 58.16 | 49.83 | 29.59 | 29.59 | 55.86 | 59.79 |
| | | CLAP | 60.64 | 59.17 | 65.14 | 62.17 | 75.37 | 71.76 | 68.82 | 56.28 | 39.70 | 37.69 | 44.12 | 41.72 |
| | | KNNCon | 65.87 | 66.21 | 83.65 | 79.75 | 69.59 | 68.30 | 62.97 | 55.10 | 56.32 | 56.45 | 42.44 | 46.51 |
| | | DyEn | 67.84 | 65.94 | 82.12 | 77.41 | 52.79 | 59.46 | 41.06 | 44.12 | 61.98 | 59.29 | 28.22 | 37.47 |
| | LLaMA | LLM-OOD | 73.97 | 66.33 | 65.66 | 66.68 | 86.65 | 80.56 | 80.03 | 61.60 | 69.86 | 62.26 | 73.83 | 68.15 |
| | | OpenMax | 84.14 | 64.24 | _89.38_ | 80.16 | 33.31 | 39.94 | 26.55 | 38.05 | 35.35 | 42.10 | 22.13 | 26.62 |
| | | EnergyBased | _86.02_ | 72.05 | 88.93 | _83.32_ | 91.60 | 83.19 | 83.19 | 51.40 | 76.17 | 64.27 | 87.25 | 69.39 |
| | | VOS | 85.08 | _75.16_ | 85.54 | 62.55 | 83.98 | 66.65 | 82.95 | _63.49_ | 77.76 | 65.17 | 83.31 | 63.37 |
| | | ViM | 78.83 | 50.82 | 86.50 | 72.91 | 89.13 | 75.26 | 78.05 | 62.37 | 73.99 | 67.21 | 87.61 | _75.96_ |
| | | SHE | 58.68 | 52.11 | 75.83 | 67.76 | 40.83 | 44.28 | 48.91 | 47.99 | 46.69 | 51.05 | 22.32 | 27.12 |
| | | TemperatureScaling | 45.24 | 50.57 | 45.49 | 54.15 | 38.96 | 43.71 | 44.69 | 45.53 | 37.11 | 43.55 | 50.14 | 46.05 |
| | | LogitNorm | 71.73 | 66.34 | 82.84 | 78.28 | 48.41 | 49.53 | 49.26 | 47.19 | 30.31 | 38.87 | 45.87 | 43.36 |
| | | NPO | 84.87 | 74.63 | 82.74 | 82.81 | _93.29_ | _87.61_ | _83.56_ | 56.98 | 78.55 | 63.57 | _87.83_ | 68.99 |
| | | UnLLM | **87.86** | **75.89** | **90.58** | **84.16** | **94.00** | **89.87** | **86.98** | **64.44** | **86.75** | **72.24** | **91.66** | **86.34** |
| 0.5 | CNN | DOC | 75.94 | 74.79 | 84.00 | 84.23 | 78.17 | 77.86 | 61.84 | 56.25 | 75.97 | 77.46 | 58.97 | 49.16 |
| | LSTM | DeepUnk | 67.67 | 60.14 | 76.36 | 69.96 | 79.65 | 77.26 | 34.80 | 40.87 | 65.53 | 68.28 | 50.01 | 57.81 |
| | BERT | OpenMax | 57.42 | 44.16 | 62.53 | 53.81 | 55.47 | 66.38 | 45.84 | 54.83 | 56.52 | 64.78 | 58.78 | 69.58 |
| | | Energy | 72.24 | 78.13 | 79.83 | 85.74 | 72.68 | 77.56 | 48.49 | 59.02 | 68.49 | 74.57 | 62.78 | 69.61 |
| | | VOS | 54.17 | 51.20 | 63.82 | 72.31 | 84.27 | 84.17 | 65.61 | 50.14 | 62.64 | 64.39 | 68.18 | 75.49 |
| | | ViM | 63.17 | 65.92 | 67.61 | 77.34 | 79.53 | 81.96 | 64.77 | 62.27 | 61.36 | 67.84 | 74.81 | 79.10 |
| | | SHE | 61.54 | 72.65 | 65.99 | 78.58 | 63.91 | 72.32 | 54.08 | 61.99 | 58.56 | 68.15 | 61.32 | 70.52 |
| | | TemperatureScaling | 58.54 | 47.39 | 67.26 | 68.39 | 65.43 | 72.50 | 58.33 | 57.27 | 54.38 | 59.03 | 54.67 | 66.95 |
| | | LogitNorm | 55.60 | 38.39 | 62.53 | 36.82 | 77.16 | 79.38 | 59.54 | 42.46 | 54.24 | 44.33 | 58.68 | 69.59 |
| | | NPO | 56.14 | 44.04 | 58.90 | 70.84 | 85.38 | 84.92 | 66.23 | 51.12 | 61.70 | 56.13 | 67.66 | 75.16 |
| | | ADB | 59.13 | 62.93 | 65.75 | 71.02 | 83.66 | 84.54 | 48.38 | 51.47 | 28.96 | 32.90 | 72.89 | 75.82 |
| | | CLAP | 53.55 | 55.99 | 64.31 | 67.11 | 84.72 | 85.02 | 53.91 | 55.03 | 46.07 | 49.54 | 66.28 | 70.37 |
| | | KNNCon | 76.54 | 80.21 | 73.10 | 82.44 | 83.54 | 84.93 | 54.47 | 60.73 | 71.38 | 76.33 | 66.41 | 72.95 |
| | | DyEn | 72.69 | 78.29 | 84.34 | 88.30 | 68.47 | 75.53 | 47.78 | 59.21 | 71.16 | 76.40 | 58.97 | 67.15 |
| | LLaMA | LLM-OOD | 79.39 | 75.23 | 64.92 | 78.42 | 89.43 | 89.70 | 61.80 | 65.29 | 75.51 | 76.74 | 80.88 | _82.80_ |
| | | OpenMax | 72.41 | 64.22 | 86.94 | 85.34 | 70.52 | 75.89 | 44.53 | 57.89 | 67.18 | 73.15 | 53.87 | 66.12 |
| | | EnergyBased | 79.78 | 75.16 | 89.76 | 88.42 | 90.06 | 89.67 | 67.44 | 48.38 | 75.98 | 75.52 | _82.00_ | 80.25 |
| | | VOS | 81.74 | 79.06 | 76.67 | 66.57 | 70.74 | 60.47 | _71.93_ | 62.30 | 75.80 | 76.71 | 78.97 | 78.08 |
| | | ViM | 66.11 | 49.26 | 83.52 | 78.36 | 88.41 | 87.23 | 68.54 | _67.68_ | _77.60_ | _78.94_ | 81.59 | 82.60 |
| | | SHE | 66.38 | 69.64 | 79.90 | 84.85 | 77.48 | 80.72 | 54.37 | 62.08 | 66.26 | 72.36 | 57.25 | 67.79 |
| | | TemperatureScaling | 60.52 | 72.17 | 64.57 | 78.09 | 62.80 | 70.27 | 55.38 | 62.82 | 58.03 | 66.79 | 57.63 | 68.66 |
| | | LogitNorm | 80.96 | _80.90_ | 90.34 | _91.92_ | 69.74 | 75.18 | 66.30 | 66.48 | 61.72 | 69.38 | 61.81 | 71.21 |
| | | NPO | _81.86_ | 78.84 | 91.31 | 91.31 | 90.38 | 90.20 | 71.20 | 58.95 | 77.16 | 78.04 | 81.04 | 80.15 |
| | | UnLLM | **84.57** | **82.82** | **92.96** | **93.42** | **92.20** | **91.97** | **76.32** | **69.26** | **80.35** | **83.34** | **82.49** | **84.32** |
| 0.75 | CNN | DOC | 73.33 | 78.68 | 83.24 | 87.30 | 81.40 | 84.55 | 55.78 | 54.17 | 67.51 | 68.08 | 36.87 | 27.97 |
| | LSTM | DeepUnk | 60.50 | 59.22 | 63.92 | 58.65 | 77.96 | 81.72 | 29.48 | 26.82 | 63.44 | 64.13 | 63.08 | 67.05 |
| | BERT | OpenMax | 55.86 | 57.50 | 65.84 | 69.09 | 70.95 | 76.60 | 52.83 | 58.36 | 69.72 | 74.05 | 73.60 | 79.20 |
| | | Energy | 77.04 | 85.27 | 87.74 | 92.70 | 83.49 | 86.82 | 56.99 | 63.34 | 78.21 | 82.83 | 80.27 | 83.37 |
| | | VOS | 56.00 | 58.05 | 75.57 | 80.31 | 82.94 | 86.62 | 50.66 | 48.46 | 66.26 | 69.15 | 81.04 | 84.53 |
| | | ViM | 69.60 | 74.58 | 79.64 | 86.61 | 84.50 | 87.40 | 57.34 | 58.67 | 72.96 | 76.94 | 83.94 | 86.55 |
| | | SHE | 74.58 | 84.04 | 83.48 | 90.85 | 78.97 | 83.56 | 59.89 | 65.81 | 75.97 | 80.89 | 77.54 | 82.37 |
| | | TemperatureScaling | 53.86 | 53.40 | 74.07 | 79.85 | 78.08 | 82.70 | 53.15 | 54.93 | 63.08 | 65.45 | 73.37 | 79.00 |
| | | LogitNorm | 39.48 | 28.06 | 43.88 | 31.32 | 79.90 | 84.04 | 45.34 | 39.24 | 54.84 | 50.82 | 76.27 | 81.22 |
| | | NPO | 55.22 | 56.15 | 75.80 | 82.62 | 84.21 | 87.22 | 50.80 | 48.85 | 65.88 | 69.06 | 81.89 | 84.83 |
| | | ADB | 65.28 | 71.57 | 71.90 | 78.17 | 83.73 | 86.87 | 43.81 | 47.05 | 30.83 | 34.00 | 72.59 | 77.51 |
| | | CLAP | 57.02 | 62.10 | 65.13 | 70.98 | 84.43 | 87.46 | 50.33 | 52.73 | 40.82 | 44.55 | 66.10 | 70.83 |
| | | KNNCon | _80.19_ | 86.46 | 90.17 | 93.80 | 85.20 | 88.46 | 60.64 | 64.80 | _80.80_ | _84.87_ | _85.39_ | _88.14_ |
| | | DyEn | 79.13 | 86.24 | 89.16 | 93.25 | 78.52 | 83.47 | 57.12 | 63.15 | 78.26 | 82.85 | 79.94 | 82.95 |
| | LLaMA | LLM-OOD | 70.27 | 71.93 | 84.69 | 91.68 | 85.72 | 87.92 | 58.24 | 64.34 | 79.48 | 82.27 | 85.02 | 87.96 |
| | | OpenMax | 65.94 | 70.63 | 83.21 | 87.25 | 82.59 | 87.09 | 55.72 | 62.13 | 77.61 | 81.13 | 70.41 | 76.52 |
| | | EnergyBased | 72.70 | 76.62 | 87.20 | 90.08 | 82.45 | 85.20 | 41.59 | 31.92 | 67.04 | 69.90 | 77.12 | 79.48 |
| | | VOS | 75.25 | 79.44 | 81.33 | 84.53 | 78.79 | 82.56 | 51.85 | 48.84 | 71.13 | 74.69 | 80.52 | 82.96 |
| | | ViM | 41.23 | 31.52 | 63.62 | 62.56 | 76.23 | 78.81 | 59.10 | 59.30 | 75.99 | 79.11 | 83.80 | 85.18 |
| | | SHE | 63.39 | 70.09 | 84.17 | 87.73 | 82.14 | 86.49 | 60.74 | 60.06 | 73.47 | 77.95 | 75.06 | 79.74 |
| | | TemperatureScaling | 75.76 | 84.08 | 81.20 | 90.07 | 78.56 | 84.34 | 60.57 | 65.72 | 74.08 | 79.06 | 74.27 | 79.60 |
| | | LogitNorm | 74.01 | 77.31 | _91.48_ | _94.11_ | 84.29 | 88.41 | 60.16 | 62.01 | 76.90 | 81.06 | 76.13 | 81.07 |
| | | NPO | 75.43 | 79.64 | 90.48 | 93.15 | _86.87_ | _89.42_ | 46.11 | 39.27 | 72.34 | 75.66 | 76.20 | 77.92 |
| | | UnLLM | **84.54** | **88.31** | **94.87** | **96.88** | **89.68** | **91.78** | **62.16** | **67.48** | **85.38** | **88.00** | **90.26** | **92.26** |

Table S9: Performance of various methods across 6 datasets at different ratios. Metrics include K-F1 and N-F1. The best results are highlighted in bold, while the second-best results are underscored. Each result represents the mean value of four repetitive experiments.

| Ratio | Backbone | Method | BANKING K-F1 | N-F1 | CLINC K-F1 | N-F1 | StackOverflow K-F1 | N-F1 | Reviews K-F1 | N-F1 | Newsgroups K-F1 | N-F1 | THUCnews K-F1 | N-F1 |
|---|---|---|---|---|---|---|---|---|---|---|---|---|---|---|
| | CNN | DOC | 66.40 | 81.57 | 75.46 | 85.63 | 69.81 | 83.61 | 59.59 | 79.57 | 66.21 | 89.59 | 56.43 | 82.27 |
| | LSTM | DeepUnk | 70.87 | 87.74 | 75.31 | 91.36 | 67.10 | 88.93 | 45.18 | 51.59 | 56.19 | 71.54 | 41.40 | 46.72 |
| 0.25 | BERT | OpenMax | 32.05 | 65.29 | 32.89 | 70.50 | 47.87 | 20.46 | 39.48 | 23.55 | 44.18 | 20.41 | 47.18 | 51.10 |
| | | Energy | 57.00 | 65.69 | 71.41 | 79.53 | 61.59 | 65.84 | 42.54 | 38.12 | 57.43 | 66.66 | 48.26 | 38.90 |
| | | VOS | 36.72 | 42.86 | 54.07 | 53.21 | 65.03 | 82.17 | 49.03 | 82.08 | 48.99 | 53.49 | 57.58 | 75.74 |
| | | ViM | 51.95 | 58.11 | 61.13 | 65.44 | 56.58 | 57.45 | 49.05 | 63.86 | 48.03 | 40.52 | 64.06 | 82.02 |
| | | SHE | 53.10 | 40.21 | 56.13 | 48.53 | 48.92 | 28.95 | 43.72 | 38.71 | 48.58 | 34.37 | 46.60 | 55.77 |
| | | TemperatureScaling | 36.24 | 60.34 | 53.52 | 57.35 | 51.01 | 38.50 | 41.98 | 56.37 | 42.68 | 28.16 | 41.06 | 36.77 |
| | | LogitNorm | 18.57 | 81.24 | 44.94 | 82.25 | 55.88 | 55.25 | 37.08 | 68.51 | 43.79 | 47.43 | 35.85 | 5.68 |
| | | NPO | 36.70 | 48.96 | 56.38 | 59.09 | 63.90 | 82.72 | 46.18 | 83.70 | 49.83 | 55.63 | 58.57 | 76.24 |
| | | ADB | 51.07 | 61.84 | 59.00 | 66.67 | 75.91 | 88.37 | 48.27 | 68.56 | 28.70 | 34.02 | 58.98 | 62.20 |
| | | CLAP | 58.69 | 68.21 | 61.88 | 72.75 | 69.75 | 81.81 | 54.45 | 78.20 | 35.79 | 47.22 | 39.23 | 49.17 |
| | | KNNCon | 65.97 | 70.63 | 79.53 | 87.85 | 66.83 | 75.63 | 53.75 | 71.26 | 55.71 | 60.18 | 47.91 | 42.33 |
| | | DyEn | 65.53 | 73.71 | 77.16 | 86.85 | 60.17 | 55.88 | 44.28 | 55.88 | 57.74 | 67.04 | 43.99 | 17.90 |
| | LLaMA | LLM-OOD | 65.62 | 79.91 | 66.65 | 67.89 | 78.54 | 90.67 | 60.09 | 80.68 | 59.33 | 76.93 | 64.17 | 80.08 |
| | | OpenMax | 62.89 | 89.85 | 79.82 | 92.38 | 43.70 | 21.13 | 40.07 | 13.76 | 45.31 | 26.08 | 33.06 | 7.29 |
| | | EnergyBased | 71.07 | 90.82 | 83.10 | 91.46 | 80.91 | 94.57 | 48.21 | 89.69 | 60.45 | 83.35 | 61.72 | 92.41 |
| | | VOS | 74.39 | 89.82 | 61.79 | 90.95 | 62.06 | 89.64 | 61.35 | 89.09 | 61.28 | 84.66 | 54.59 | 89.72 |
| | | ViM | 48.95 | 86.32 | 72.41 | 91.10 | 71.67 | 93.18 | 60.41 | 86.38 | 64.53 | 80.61 | 70.41 | 92.62 |
| | | SHE | 51.41 | 65.38 | 67.38 | 81.77 | 45.82 | 36.60 | 47.52 | 53.54 | 51.84 | 47.08 | 33.53 | 7.91 |
| | | TemperatureScaling | 50.89 | 44.40 | 54.43 | 43.52 | 45.94 | 32.58 | 45.33 | 47.91 | 46.18 | 30.43 | 42.49 | 56.73 |
| | | LogitNorm | 65.73 | 77.89 | 78.04 | 87.29 | 49.79 | 48.24 | 46.59 | 54.46 | 43.48 | 15.87 | 42.66 | 45.48 |
| | | NPO | 73.83 | 89.73 | 82.60 | 90.29 | 86.01 | 95.60 | 54.25 | 89.78 | 59.19 | 85.49 | 61.06 | 92.79 |
| | | UnLLM | 75.04 | 92.02 | 83.90 | 93.58 | 88.65 | 96.00 | 62.16 | 91.94 | 68.33 | 91.82 | 83.63 | 94.46 |
| | CNN | DOC | 74.50 | 77.92 | 84.10 | 84.38 | 77.96 | 79.57 | 55.42 | 67.52 | 77.86 | 74.67 | 46.63 | 64.58 |
| | LSTM | DeepUnk | 59.75 | 74.91 | 69.80 | 81.54 | 76.79 | 81.86 | 41.86 | 20.12 | 69.07 | 59.93 | 62.55 | 24.58 |
| 0.5 | BERT | OpenMax | 43.75 | 59.95 | 53.68 | 63.20 | 70.21 | 28.10 | 56.02 | 25.16 | 68.00 | 32.49 | 74.31 | 36.53 |
| | | Energy | 78.40 | 68.00 | 85.88 | 75.78 | 78.59 | 67.28 | 60.22 | 29.17 | 76.34 | 56.83 | 73.23 | 44.34 |
| | | VOS | 51.11 | 54.45 | 72.59 | 51.59 | 84.12 | 84.71 | 49.23 | 72.85 | 65.17 | 56.54 | 77.79 | 59.34 |
| | | ViM | 66.06 | 60.41 | 77.62 | 56.46 | 82.45 | 77.03 | 62.07 | 67.11 | 70.06 | 45.74 | 80.42 | 69.88 |
| | | SHE | 73.35 | 46.27 | 78.96 | 50.20 | 74.45 | 50.95 | 62.72 | 43.72 | 71.87 | 30.95 | 74.51 | 42.59 |
| | | TemperatureScaling | 46.96 | 63.64 | 68.45 | 63.35 | 74.06 | 56.95 | 57.14 | 60.47 | 61.01 | 39.22 | 72.74 | 26.41 |
| | | LogitNorm | 37.74 | 63.08 | 36.37 | 70.60 | 79.72 | 76.01 | 41.48 | 67.04 | 43.52 | 52.43 | 74.08 | 38.14 |
| | | NPO | 43.58 | 61.90 | 71.23 | 41.63 | 84.82 | 85.99 | 50.23 | 73.42 | 55.76 | 59.82 | 77.56 | 58.33 |
| | | ADB | 63.09 | 56.92 | 71.14 | 61.67 | 83.95 | 83.41 | 51.56 | 49.18 | 33.99 | 22.00 | 76.67 | 69.88 |
| | | CLAP | 56.04 | 54.13 | 67.15 | 63.94 | 84.99 | 85.36 | 55.00 | 55.79 | 50.37 | 41.19 | 71.49 | 62.51 |
| | | KNNCon | 80.35 | 75.13 | 82.71 | 62.30 | 85.11 | 83.12 | 61.31 | 46.13 | 77.60 | 63.58 | 76.44 | 48.49 |
| | | DyEn | 78.55 | 68.34 | 88.38 | 82.46 | 77.06 | 60.25 | 60.47 | 27.64 | 77.83 | 62.19 | 71.79 | 34.68 |
| | LLaMA | LLM-OOD | 75.06 | 81.65 | 78.85 | 46.07 | 89.71 | 89.54 | 65.55 | 59.18 | 77.11 | 72.95 | 83.34 | 78.66 |
| | | OpenMax | 63.87 | 77.49 | 85.31 | 88.05 | 77.94 | 55.37 | 59.56 | 16.11 | 74.99 | 54.82 | 72.34 | 22.59 |
| | | EnergyBased | 74.96 | 82.76 | 88.39 | 90.50 | 89.58 | 90.57 | 47.32 | 74.93 | 75.39 | 76.77 | 82.11 | 80.26 |
| | | VOS | 78.93 | 83.94 | 66.38 | 80.99 | 58.88 | 76.33 | 61.71 | 77.04 | 76.88 | 74.99 | 77.78 | 80.22 |
| | | ViM | 48.61 | 73.91 | 78.26 | 85.71 | 87.01 | 89.37 | 67.51 | 72.05 | 79.14 | 76.92 | 84.09 | 80.21 |
| | | SHE | 69.77 | 64.48 | 84.96 | 76.58 | 81.49 | 72.99 | 62.94 | 40.44 | 74.07 | 55.31 | 72.65 | 33.80 |
| | | TemperatureScaling | 72.90 | 44.64 | 78.51 | 46.01 | 72.79 | 45.06 | 63.47 | 46.54 | 70.23 | 32.41 | 73.52 | 34.69 |
| | | LogitNorm | 80.88 | 82.01 | 91.95 | 89.94 | 76.84 | 58.60 | 66.44 | 67.40 | 72.05 | 42.74 | 74.94 | 45.14 |
| | | NPO | 78.70 | 84.12 | 91.31 | 91.53 | 90.14 | 90.78 | 58.23 | 76.82 | 78.17 | 76.73 | 82.37 | 79.90 |
| | | UnLLM | 82.74 | 85.72 | 93.42 | 93.00 | 91.91 | 92.60 | 68.81 | 80.59 | 83.95 | 77.20 | 84.83 | 80.75 |
| | CNN | DOC | 78.85 | 61.25 | 87.37 | 72.81 | 84.90 | 72.30 | 53.95 | 52.53 | 67.67 | 57.58 | 29.77 | 41.32 |
| | LSTM | DeepUnk | 59.27 | 54.85 | 58.57 | 67.62 | 82.57 | 68.93 | 27.31 | 11.81 | 65.95 | 40.93 | 70.76 | 26.25 |
| 0.75 | BERT | OpenMax | 57.82 | 39.45 | 69.22 | 53.91 | 79.52 | 32.82 | 59.38 | 19.81 | 77.22 | 26.52 | 83.38 | 37.46 |
| | | Energy | 85.93 | 47.26 | 92.87 | 73.63 | 87.99 | 69.29 | 64.52 | 18.80 | 85.25 | 46.46 | 88.15 | 30.87 |
| | | VOS | 58.38 | 39.45 | 80.51 | 58.20 | 87.59 | 72.19 | 48.49 | 47.16 | 70.65 | 46.54 | 86.14 | 68.39 |
| | | ViM | 75.04 | 48.39 | 86.89 | 54.43 | 87.34 | 74.36 | 58.90 | 50.25 | 78.88 | 47.80 | 87.76 | 74.51 |
| | | SHE | 84.96 | 31.57 | 91.15 | 56.93 | 85.43 | 55.57 | 66.51 | 38.91 | 84.06 | 33.25 | 85.32 | 52.86 |
| | | TemperatureScaling | 53.62 | 40.88 | 80.09 | 52.40 | 84.23 | 59.70 | 55.19 | 45.08 | 67.31 | 37.62 | 82.57 | 43.38 |
| | | LogitNorm | 27.83 | 40.80 | 31.19 | 45.15 | 85.28 | 65.41 | 45.33 | 43.79 | 51.27 | 44.17 | 83.83 | 55.11 |
| | | NPO | 56.45 | 39.05 | 82.92 | 49.72 | 88.02 | 75.09 | 48.89 | 47.05 | 70.67 | 44.89 | 86.14 | 71.73 |
| | | ADB | 72.07 | 43.48 | 78.40 | 51.69 | 87.65 | 75.10 | 47.35 | 35.55 | 35.07 | 17.93 | 79.59 | 54.59 |
| | | CLAP | 62.48 | 40.37 | 71.19 | 47.04 | 88.23 | 75.98 | 53.00 | 42.43 | 46.16 | 20.47 | 72.82 | 50.93 |
| | | KNNCon | 86.90 | 61.61 | 93.91 | 80.67 | 89.38 | 74.55 | 65.22 | 49.02 | 86.36 | 62.53 | 90.49 | 63.10 |
| | | DyEn | 86.79 | 54.74 | 93.38 | 78.02 | 85.43 | 54.09 | 64.24 | 21.81 | 85.24 | 47.06 | 87.88 | 28.73 |
| | LLaMA | LLM-OOD | 72.10 | 62.34 | 91.96 | 60.82 | 88.54 | 77.79 | 65.02 | 38.54 | 83.29 | 67.06 | 89.41 | 71.94 |
| | | OpenMax | 70.88 | 56.18 | 87.37 | 73.66 | 88.69 | 63.09 | 63.49 | 10.57 | 83.14 | 51.02 | 81.39 | 27.75 |
| | | EnergyBased | 76.84 | 63.80 | 90.18 | 79.11 | 85.94 | 74.07 | 31.57 | 44.87 | 70.78 | 56.69 | 80.33 | 71.01 |
| | | VOS | 79.68 | 65.65 | 84.64 | 72.68 | 83.54 | 67.85 | 48.83 | 49.18 | 75.72 | 59.23 | 83.85 | 74.05 |
| | | ViM | 31.27 | 45.94 | 62.59 | 58.71 | 79.51 | 68.31 | 59.37 | 56.56 | 80.12 | 63.99 | 86.40 | 78.06 |
| | | SHE | 70.70 | 35.08 | 87.87 | 71.67 | 88.19 | 60.97 | 67.09 | 55.85 | 79.80 | 50.14 | 82.78 | 49.35 |
| | | TemperatureScaling | 84.85 | 39.79 | 90.47 | 45.20 | 87.03 | 43.91 | 66.34 | 42.15 | 81.98 | 35.29 | 82.94 | 46.18 |
| | | LogitNorm | 77.53 | 64.82 | 94.20 | 84.11 | 89.83 | 67.13 | 62.29 | 51.37 | 82.70 | 56.41 | 83.73 | 54.42 |
| | | NPO | 79.89 | 65.79 | 93.24 | 83.14 | 90.10 | 79.22 | 39.08 | 46.74 | 76.68 | 60.36 | 78.67 | 70.35 |
| | | UnLLM | 88.58 | 72.60 | 96.94 | 89.73 | 92.36 | 83.08 | 67.84 | 56.65 | 89.06 | 72.18 | 93.55 | 78.11 |

