# OpenReview forum: "Beyond the Known: An Unknown-Aware Large Language Model for Open-Set Text Classification"
_ICLR.cc/2026/Conference — ICLR 2026 Poster_

### Official Review · Reviewer_rWfQ · 2025-10-19

**Soundness:** 3
**Presentation:** 3
**Contribution:** 3
**Rating:** 6
**Confidence:** 4

**Summary:**

The paper tackles open-set text classification (OSTC) by making “unknown-class” awareness an explicit, trainable behavior for LLMs via UnLLM, a three-stage pipeline.

**Strengths:**

- The three-stage pipeline (contrastive + orthogonality regularization; K+1 logit calibration; analogy-augmented inference) is coherent and practical.

- Shows consistent improvements on several benchmarks.

- Simple, implementable training recipe that may make LLM classifiers less overconfident under partial label contexts.

- The paper does not over-promise formal guarantees; instead it presents precise objectives/losses and a closed-form calibration projection (Eq. 4) with clear tensor-shape intent.

**Weaknesses:**

- This paper feels close in spirit to prior OE extensions and recent fine-tuning-with-auxiliary-data methods (WOODS, VOS, NPOS).

**Questions:**

- Consider a compact numeric ablation table ****(mean±std) and a reliability diagram (ECE).

- What encoder, k, and index size are used for analogy; what’s the per-example latency?

- Nice motivation—could you add one controlled test that separates *withheld in-domain labels* (conditional-OOD) from *cross-domain OOD*, to empirically reinforce generalization.

- The experimental section should take into account advanced OOD detection methods that leverage additional auxiliary OOD data, as referenced in [1, 2, 3, 4].



[1]: Ming, Yifei, et al. "Poem: Out-of-distribution detection with posterior sampling." *International Conference on Machine Learning*. PMLR, 2022.

[2]: Wang, Qizhou, et al. "Out-of-distribution detection with implicit outlier transformation." *arXiv preprint arXiv:2303.05033* (2023).

[3]: Zheng, Haotian, et al. "Out-of-distribution Detection Learning with Unreliable Out-of-distribution Sources." *arXiv preprint arXiv:2311.03236* (2023).

[4]: Wang, Qizhou, et al. "Learning to augment distributions for out-of-distribution detection." *arXiv preprint arXiv:2311.01796* (2023).

---

> ### Author Response · Authors · 2025-11-19
> **Response to Reviewer rWfQ**
>
> We sincerely thank the your valuable feedback. In the following, we address each concern weakness and question by point.
>
> > **W1:**“This paper feels close in spirit to prior OE extensions and recent fine-tuning-with-auxiliary-data methods (WOODS, VOS, NPOS).”
> >
>
> **Responses:**
>
> Our method differs fundamentally from WOODS, VOS, and NPOS.
> 1.	VOS and NPOS construct **virtual OOD samples** and train the model with an additional binary ID/OOD loss. In contrast, our method partitions the dataset into subsets to construct **real labeled OOD samples**, which are directly used for (K+1)-class classification.
> 2.	WOODS leverages unlabeled “in-the-wild” data, whereas **our work does not use any external data** beyond the original training set.
>
> We will expand the discussion of WOODS and other OE-style methods that depend on additional OOD datasets in the Related Work section of the revision.
>
> > **Q1:**“What encoder, k, and index size are used for analogy; what’s the per-example latency?”
> >
>
> **Responses:**
>
> As stated in **Line 941**, we use the BERT pooler layer as the encoder for analogy retrieval, with all embeddings pre-computed before inference. We set k = 20 (top-20 nearest neighbors), and the index size corresponds to the entire training set.
>
> Regarding latency, **Line 1174** already reports the empirical measurement. On the CLINC dataset (2,205 samples), using 5-thread inference on a single NVIDIA 5880 GPU, inference time increased only from 8m59s (without analogy) to 10m46s (with analogy), i.e., ~20% per-example overhead. This modest increase reflects the relatively small analogy prompt (less than 20% of the total input length) and demonstrates that the retrieval + prompting mechanism introduces only limited latency while providing consistent performance and interpretability gains.
>
> > **Q2:**“Nice motivation—could you add one controlled test that separates withheld in-domain labels (conditional-OOD) from cross-domain OOD, to empirically reinforce generalization.”
> >
>
>
> **Responses:**
> The scenario you described does not align with the formal definition of the OSTC problem. OSTC assumes access to labeled data from only the K in-domain classes, from which we aim to train an (K + 1)-class classifier. This formulation is highly relevant in realistic, dynamically evolving environments where obtaining external data is often difficult. Precisely because no OOD data—or any additional external data—is available during training, the task becomes challenging. **A substantial body of prior work has adopted this setting and conducted research under the same constraints [1,2,3,4,5]**.
>
> Under this formulation, **only in-domain labels are accessible during training**, while all other labels are strictly unavailable. For this reason, the controlled comparison you suggested—explicitly separating withheld in-domain labels (conditional-OOD) from cross-domain OOD—falls outside the scope of this setting.
>
> > **Q3:**“The experimental section should take into account advanced OOD detection methods that leverage additional auxiliary OOD data.”
> >
>
> **Responses:**
>
> We would like to emphasize that our task is fundamentally an OSTC problem, where the model is trained only on ID classes, and **no external OOD data is permitted** during training. This restriction is inherent to the problem itself, and also the primary challenge: the model must identify and separate unknown classes without ever seeing OOD examples.
>
> Therefore, advanced OOD detection methods that rely on auxiliary OOD datasets, external negative samples, or synthetic OOD generation fall outside the scope of our setting and **cannot be applied fairly or directly**.
>
> **Refs:**
>
> [1] Lei Shu, Hu Xu, and Bing Liu. DOC: Deep Open Classification of Text Documents. EMNLP, 2017.
> [2] Sridhama Prakhya, Vinodini Venkataram, and Jugal Kalita. Open set text classification using CNNs. ICON, 2017.
> [3] Hanlei Zhang, Hua Xu, and Ting-En Lin. Deep open intent classification with adaptive decision boundary. AAAI, 2021.
> [4] Yunhua Zhou, Peiju Liu, and Xipeng Qiu. KNN-contrastive learning for out-of-domain intent classification. ACL, 2022.
> [5] Yunhua Zhou, Jianqiang Yang, Pengyu Wang, and Xipeng Qiu. Two birds one stone: Dynamic ensemble for OOD intent classification. ACL, 2023.

---

### Official Review · Reviewer_163y · 2025-10-20

**Soundness:** 1
**Presentation:** 2
**Contribution:** 2
**Rating:** 2
**Confidence:** 3

**Summary:**

This paper investigates the open-set text classification (OSTC) problem, in which models must correctly classify in-distribution (ID) samples while identifying out-of-distribution (OOD) inputs. The authors propose UnLLM, an unknown-aware large language model that integrates open-set awareness directly into the fine-tuning process. The method reformulates classification as a subset-conditioned generation task, where random subsets of known labels are provided and instances whose true labels are excluded are treated as 'unknown'. To enhance representation quality and calibration, UnLLM incorporates contrastive learning, orthogonality constraints, and a post-hoc logit calibration step that aligns output weights with OOD-relevant representations. During inference, an analogy-augmented self-reflection mechanism enables the model to reassess uncertain cases and mitigate overconfidence. Experiments on six benchmark datasets demonstrate that UnLLM consistently outperforms both traditional PLM-based methods (e.g. ADB, CLAP, KNNCon) and recent LLM-based baselines (LLM-OOD, VOS, NPO), achieving strong results in both ID classification (K-F1) and OOD detection (N-F1).

**Strengths:**

* The main contribution of the paper lies in reformulating OSTC as a subset-conditioned text generation task, extending prior work that handled OOD detection only as a post-hoc step. This framing is new in the context of large language models and conceptually straightforward.

* The authors evaluate their approach on six benchmark datasets and a broad range of baselines, covering both traditional PLM-based and more recent LLM-based methods. The experimental coverage is thorough and shows consistent empirical improvements.

* Although the design is primarily empirical, the results suggest that the proposed paradigm can be effectively applied to large models and yields measurable performance gains in OSTC scenarios.

**Weaknesses:**

The main issue of the paper lies in its limited scientific depth. While the proposed framework achieves promising empirical results, it lacks sufficient theoretical or empirical analysis to explain why existing methods fail and why the proposed reformulation, which frames OSTC as a subset-conditioned text generation task, is essential or fundamentally different. As a result, the paper does not convincingly establish the underlying rationale or mechanisms that support its claimed contributions. The main concerns are summarised as follows:

* **Lack of clear evidence that the proposed reformulation is essential.**
  The paper provides weak evidence that the subset-conditioned text generation reformulation is fundamentally important for open-set text classification. While the full framework shows improvements, the comparison setup is confusing and does not isolate the benefit of the reformulation itself.

  * In particular, the representation-learning components (contrastive learning and orthogonality constraints) can be readily applied to existing methods, and their effectiveness has already been demonstrated in prior work. As shown in Table 1 and Figure 4, removing these components (‘w/o CL’ or ‘w/o Orth’) significantly reduces performance, often below other LLM-based baselines, suggesting that most of the gains may stem from these well-known techniques rather than from the proposed reformulation.
  * The Reflective Inference (Analogy-Augmented Self-Reflection) module also appears largely independent of the new formulation. It is introduced as an inference-time enhancement and could be attached to any model trained under different paradigms. Since this component does not interact with the subset-conditioned fine-tuning objective, its inclusion further obscures the specific contribution of the reformulation itself.
  * To substantiate the importance of the new formulation, the authors should include **control experiments** to isolate its contribution more clearly. *For example*, they could apply the same contrastive, orthogonality, and reflection modules to standard generative fine-tuning or other LLM-based baselines (such as LLM-OOD or VOS) without the subset-conditioning step. Comparing such models with and without access to the subset of known labels would help determine whether the reformulation itself contributes meaningfully beyond these auxiliary components.
  * Additionally, while the logit calibration component may depend on the new formulation, the paper also lacks a clear, controlled experimental analysis identifying which parts of the framework genuinely benefit from it.


* **Unfair and inconclusive comparison between LLM-based and BERT-based baselines.**
  The paper emphasises that one of its main contributions is the application of LLMs to OSTC and reports that LLM-based baselines outperform BERT-based methods (Section 5.2). However, this comparison is not meaningful, as the models differ drastically in scale and architecture: the LLaMA-3.1-8B model contains billions of parameters, whereas the BERT-base models are two orders of magnitude smaller. The observed performance gap therefore likely reflects differences in model capacity and pretraining rather than an inherent advantage of the proposed generative fine-tuning paradigm.

* **Lack of evidence for consistency across model backbones.**
  The paper reports results on two LLM architectures (LLaMA 3.1-8B and Qwen 2.5-7B), but the baselines are only evaluated on LLaMA. As a result, it is unclear whether the proposed UnLLM framework consistently improves performance across different backbones or whether the observed gains are specific to a single model. To claim general applicability, the authors should include baseline comparisons on Qwen 2.5-7B or at least provide an analysis showing that the observed improvements are robust to architectural variations.

* **Unclear and unsupported motivation, both theoretically and empirically.**
  The paper’s motivation is difficult to follow at both the conceptual and empirical levels. Several key statements, such as 'LLMs often exhibit overconfidence in their predictions' (Section 4.3) and 'we observe a misalignment between the LLM’s internal knowledge (representation space) and its outputs' (Section 4.2), are presented as empirical observations but are not supported by any systematic study or quantitative evidence. It remains unclear whether these issues actually occur under the presented OSTC setting, and no diagnostic analysis is provided to demonstrate their impact. Furthermore, the paper does not establish a clear theoretical or conceptual connection between these claimed problems and the proposed solutions. As a result, it is difficult to assess whether the introduced components are sufficient or even relevant to address the stated challenges.

**Questions:**

* At line 246, the authors state that the orthogonality loss minimises the projection of ID representations onto the principal subspace of outliers (Section 4.1, Eq. 8). However, minimising this projection only reduces the overlap between the two subspaces; it does not guarantee orthogonality or decorrelation in a strict sense. Could the authors clarify the theoretical rationale behind this formulation? Why is minimising the projection considered sufficient to ensure separation, rather than explicitly enlarging the null space of the outlier subspace or directly enforcing an orthogonality constraint between the ID and OOD representation spaces?

---

> ### Author Response · Authors · 2025-11-19
> **Response to Reviewer 163y[1/6]**
>
> We sincerely thank you for your valuable comments. We believe that some of the concerns may arise from a misunderstanding of our task formulation and methodological motivation. Before addressing your specific points, we would therefore like to clarify the core objective and novelty of our work.
>
> Conventional OSTC approaches are predominantly based on closed-set training, where OOD detection is performed as a post-hoc step that rejects low-confidence predictions as OOD. However, our visualization of the learned representations shows that, under this training paradigm, the ID and OOD samples exhibit severe overlap, together with substantial overconfidence on OOD inputs. **This ill-posed representation space imposes an intrinsic performance bottleneck on any post-hoc OOD detection**. To address this, we propose a completely novel open-set fine-tuning paradigm for LLMs: by inserting a subset label set into the prompt, we reformulate the objective from maximizing $P(y \mid x)$ to maximizing $P(\tilde{y} \mid x, \mathcal{Y}^p)$. In doing so, we explicitly introduce $\tilde{y} = K+1$ when $y \notin \mathcal{Y}^p$, thereby **incorporating open-set risk awareness directly into the training process**.
>
> Building on this paradigm, we further introduce a unified optimization framework across representations, logits, and inference, which jointly tackles (i) the distribution gap, (ii) the misalignment between internal knowledge and outputs, and (iii) overconfidence on label-similar OOD samples. **All of these modules are designed under the proposed K+1 classification paradigm**, rather than being straightforward reuses of components “whose effectiveness has already been demonstrated in prior work,” as you suggested. In the following, we will respond in detail to each of the key issues you raised.

---

> ### Author Response · Authors · 2025-11-19
> **Response to Reviewer 163y[2/6]**
>
> > **W1:**“Lack of clear evidence that the proposed reformulation is essential.”
> >
>
> **Responses:**
>
> We emphasize that the proposed reformulation is indeed essential, as it provides the core mechanism that enables the effectiveness of all subsequent modules. To further substantiate this point, we present additional evidence from both an ablation study and a representation-level analysis.
>
>
> **1. Evidence from Ablation Study**：
>
> To avoid ambiguity, we clarify the two training paradigms:
> • Closed-set SFT: The LLM is fine-tuned only on ID labels, optimizing it to generate ID class outputs exclusively. No OOD semantics are introduced at training time.
> • Open-set SFT: The LLM is fine-tuned using both ID labels and conditional-OOD labels, enabling the model to explicitly learn an additional OOD class during training.
>
> We replaced the Open-set SFT with Closed-set SFT and sequentially integrated each module into both the Closed-set and Open-set pipelines for comparison. The results are shown in Table 1.
>
> The empirical results show that when modules are added on top of Closed-set SFT, the model fails to mitigate large-scale overfitting of LLM outputs to the ID label space. In fact, performance drops from an F1 of 0.48 to 0.44 (**a relative degradation of 9.48%**). Even with LLM-OOD post-hoc detection, the improvement is only marginal, reaching 0.73 F1, and still cannot overcome the inherent bottleneck.
>
> In contrast, Open-set SFT alone already yields a substantial improvement, increasing F1 from 0.48 to 0.75 (**a relative gain of 55.99%**).
> Furthermore, only when the subsequent modules are applied on top of Open-set SFT does the model reach its best performance of 0.83 F1 (**a relative gain of 90.09% and 13.32% compared with Closed-set SFT and LLM-OOD equipped with the same modules**).
>
> *Table 1: Ablation Study on CLINC dataset of 0.25 ratio.*
>
> | Method                       || ACC(%)            | F1(%)       | K-F1(%)          | N-F1(%)                |
> |------------------------------|----------------|----------------|----------|---------------|---------------------|
> |LLM Output| Closed-set SFT                           | 24.31          | 48.40    | 49.71         | 0.00                |
> || Closed-set SFT  + Contrastive Learning   | 24.53          | 48.50    | 49.80         | 0.35                |
> || Closed-set SFT  + Contrastive Learning + Orthogonal Regularizer | 24.58          | 47.53    | 48.79         | 0.82                |
> || Closed-set SFT  + Contrastive Learning + Orthogonal Regularizer + Analogy                | 36.89          | 43.81    | 43.98         | 37.27               |
> |Post-hoc Method| LLM-OOD  |   65.66  |   66.68    |       66.65  | 67.89      |
> || LLM-OOD + Contrastive Learning  |  68.00   |  30.21     |   29.71      |   78.26    |
> || LLM-OOD + Contrastive Learning + Orthogonal Regularizer  |  81.02   |  74.33     |   73.68      |    86.28    |
> || LLM-OOD + Contrastive Learning + Orthogonal Regularizer + Analogy  |  80.8   |  73.49     |   73.15      |   86.16    |
> |LLM Output| Open-set SFT  |   80.18|	75.5|	75.24|	85.3|
> || Open-set SFT  + Contrastive Learning |  82.22|	77.57|	77.32|	86.88|
> || Open-set SFT  + Contrastive Learning + Orthogonal Regularizer  |83.51|	78.98|	78.73|	88.09|
> || Open-set SFT  + Contrastive Learning + Orthogonal Regularizer + Calibaration   |85.09|	80.32|	80.09|	90.04|
> || Open-set SFT  + Contrastive Learning + Orthogonal Regularizer + Calibaration + Analogy |**89.87** |**83.28**| **83.01**| **93.15**|
>
> **2. Evidence from Representation Analysis**：
>
> Our case studies distinctly show that UnLLM produces well-separated ID and OOD representations, in sharp contrast to the heavily entangled clusters produced by Closed-set SFT.
> To further strengthen the empirical support for our claim, we additionally report Intra–Inter Ratio[4], which quantifies the degree of separation between ID and OOD samples in representation space:
>
> $$
> \text{Intra-Inter Ratio} = \frac{\text{Inter-cluster Distance}}{\text{Intra-cluster Distance}} = \frac{\text{Average Cosine Distance between ID and OOD Clusters}}{\text{Average Cosine Distance within the ID Cluster}}
> $$
>
> A higher ratio (as achieved by UnLLM) indicates compact ID clusters that are well separated from OOD samples—an essential property for robust open-set recognition. A lower ratio (as seen in Closed-set SFT) suggests significant ID–OOD overlap and poor separability.
>
> *Table 2: Performanc on Intra–Inter Ratio with different methods.*
>
> | Dataset       | Ratio | LLM-OOD (Basline) | UnLLM (Ours) |
> |---------------|----------|-------------------|--------------------|
> | StackOverflow | 0.25      | 0.99              | 1.98               |
> |  | 0.50      | 0.88              | 5.62               |
> |  | 0.75      | 0.68              | 4.56               |
> | CLINC | 0.25      | 0.90              | 1.15               |
> |  | 0.50      | 0.94              | 21.30               |
> |  | 0.75      | 1.09              | 6.43               |

---

> ### Author Response · Authors · 2025-11-19
> **Response to Reviewer 163y[3/6]**
>
> > **W1.1:**“In particular, the representation-learning components (contrastive learning and orthogonality constraints) can be readily applied to existing methods, and their effectiveness has already been demonstrated in prior work.”
> >
>
> **Responses:**
>
> We would like to clarify that **the contrastive learning used in this work is specifically designed for the open-set SFT** and carries a unique purpose in this context, while the proposed **orthogonality constraint is an entirely novel loss function**.
>
> For contrastive learning, applying it under the conventional closed-set SFT paradigm can only enlarge the distances between ID classes. As shown in Table 1, this leads to only marginal improvements (e.g., Closed-set SFT ACC **0.24→0.24**, K-F1 **0.50→0.50**) and even highly unbalanced performance for LLM-OOD (e.g., F1, **0.66→0.30**, a large drop). In contrast, under open-set SFT, our contrastive learning simultaneously enforces separation between ID and OOD classes, yielding consistent and stable gains (e.g., ACC **0.80→0.82**, F1 **0.76→0.78**), as reflected in Table 1.
>
> Regarding the orthogonality constraint, although it is loosely inspired by the Gaussian assumption in VOS and the orthogonality intuition in ViM, its nature is fundamentally different. VOS relies on post-hoc uncertainty-based score regularization, and ViM merely introduces a new post-hoc scoring method without involving training. In contrast, our method directly optimizes the model with an orthogonal loss computed on the ID and pseudo-OOD representation spaces, which has not appeared in prior work.
>
> > **W1.2:**“The Reflective Inference module also appears largely independent of the new formulation.”
> >
>
> **Responses:**
> Reflective Inference is designed to address the issue of overconfidence on samples that are semantically close to the labeled classes—i.e., confusing or near-boundary cases. Because **open-set SFT enables the model to explicitly perceive OOD risk** and internalize the notion of “OOD,” adding an **OOD-oriented reflective mechanism** at inference time becomes highly effective.
>
> As shown in Table 1, applying the reflection mechanism to LLM-OOD yields almost no improvement (e.g., F1 **0.74→0.73**). This is because closed-set training causes the model to overfit to known labels, leaving it unable to properly identify OOD instances. These results indirectly demonstrate that the effectiveness of the Reflective Inference module is tied to our proposed formulation, rather than being a standalone component.
>
> > **W1.3:**“To substantiate the importance of the new formulation, the authors should include control experiments to isolate its contribution more clearly. For example, they could apply the same contrastive, orthogonality, and reflection modules to standard generative fine-tuning or other LLM-based baselines (such as LLM-OOD or VOS) without the subset-conditioning step. Comparing such models with and without access to the subset of known labels would help determine whether the reformulation itself contributes meaningfully beyond these auxiliary components.”
> >
>
> **Responses:**
>
> To make our framework more convincing, **we designed the control experiments as shown in Table 1** and we get the key findings (CLINC dataset, 0.25 ratio):
> 1.	Closed-set SFT severely overfits to seen labels, causing **N-F1 ≈ 0** across settings.
> The model simply lacks the ability to output “unknown,” regardless of how CL or orthogonality is applied. Analogy-based reflection can partially reject OOD examples by prompting the LLM to reconsider alternatives, but due to the absence of an internal OOD boundary, its accuracy remains low. Notably, a closed-set SFT model performs worse than an un-fine-tuned LLM in open-set scenarios, indicating that closed-set training reduces the model’s inherent generalization and OOD awareness.
>
> 2.	The improvement on LLM-OOD is limited.
> (1) Adding CL substantially improves OOD detection (higher N-F1) by increasing separability in the embedding space. However, without supervised OOD anchors, contrastive learning tends to inflate the OOD region excessively, causing many ID samples to be rejected. This yields a high N-F1 (**0.68→0.78**) but a degraded K-F1 (**0.67→0.30**), leading to severe imbalance.
> (2) Adding an Orthogonal Regularizer improves stability by preventing contrastive learning from over-contracting feature representations (e.g., for LLM-OOD, F1 improves from **0.30 → 0.74**; for Open-set SFT, F1 improves from **0.79→0.80**). Its PCA-based global formulation further enhances robustness.
> (3) Analogy-based reflection provides little benefit for LLLM-OOD (**F1: 0.74→0.73**) but yields substantial gains for Open-set SFT (**F1: 0.80→0.83**). This is because LLM-OOD already rejects OOD samples using post-hoc score statistics. Adding analogy only modifies the output layer, while the underlying LLM is still fine-tuned exclusively on ID data, making its reflective reasoning for OOD inherently unreliable.

---

> ### Author Response · Authors · 2025-11-19
> **Response to Reviewer 163y[4/6]**
>
> > **W1.4:**“Additionally, while the logit calibration component may depend on the new formulation, the paper also lacks a clear, controlled experimental analysis identifying which parts of the framework genuinely benefit from it.”
> >
>
> **Responses:**
> The purpose of logit calibration is to address the persistent misalignment between an LLM’s internal knowledge and its external verbalized outputs, thereby improving the faithfulness with which internal representations are expressed through generation.
>
> To verify the presence of this misalignment in our setting, we conduct a controlled comparison across four model variants, evaluating AUC (representation separability/internal recognition) versus K-F1 / N-F1 (output correctness):
>
> • Closed-set SFT: the LLM is directly fine-tuned for generation;
> • Closed-set SFT (LLM-OOD): a post-hoc OOD detector is applied to the same model outputs;
> • Open-set SFT (no calibration): open-set training is used;
> • Open-set SFT (only calibration): open-set training with calibration applied only to the OOD token weight in the LM head.
>
> Our experiments reveal a consistent phenomenon: LLMs tend to produce affirmative, ID-biased outputs. When we extract the model’s predicted probability for the OOD token and compute AUC (OOD/ID), we observe that—even when the model’s internal ranking of ID vs. OOD samples is already highly accurate (e.g., high AUC)—its external outputs often fail to reflect this understanding and instead overwhelmingly predict ID labels.
> For example, in Table 3, Closed-set SFT achieves AUC = 0.91 but N-F1 = 0.0, demonstrating a severe disconnect between internal recognition and external expression. Open-set SFT alleviates the issue but still shows bias toward ID labels. **When we make no architectural changes and only introduce calibration on top of open-set SFT, both AUC and N-F1 improve substantially**. This confirms that calibration directly mitigates the misalignment problem rather than serving as an unrelated add-on.
>
> Importantly, calibration is only applicable to open-set SFT. **Closed-set SFT does not learn an OOD token**, nor does its validation set contain OOD samples needed to guide calibration. Thus, the component is inherently tied to—and only benefits—the new formulation.
>
>
> *Table 3: Ablation Study of Calibration Method on CLINC dataset*
>
> | Model Variant                           | 0.25   |      |  | 0.5    |     |   | 0.75    |     |   |
> |-----------------------------------------|--------|--------|--------|--------|--------    |--------|--------    |--------    |--------|
> |                                         | AUC(%)    | K-F1(%)   | N-F1(%)     | AUC(%)    | K-F1(%)    | N-F1(%)      | AUC(%)    | K-F1(%)      | N-F1(%) |
> | Closed-set SFT                  | 90.79  | 49.71  | 0.0      |  92.23 | 71.96   | 0.18      | 91.63  | 87.76     | 0.0  |
> | Closed-set SFT (LLM-OOD)        | 90.79  | 66.65  | 67.89    |  92.23 | 78.85   | 46.07     | 91.63  |  91.96    | 60.82|
> | Open-set SFT(no calibration)    | 94.12  | 77.32 |	86.88    |   97.29| 91.29   | 86.61     | 98.70  |   94.37   | 85.12|
> | Open-set SFT(only calibration)    | 96.33  | 79.3 |	88.61    |   97.47| 91.25   | 87.21     | 98.75  |  96.62   | 88.5|
> | UnLLM       | **97.29**  | **83.90**  | **93.58**    |   **98.42**| **93.42**   | **93.00**     | **99.13**  |   **96.94**   | **89.73**|

---

> ### Author Response · Authors · 2025-11-19
> **Response to Reviewer 163y[5/6]**
>
> > **W2:**“Unfair and inconclusive comparison between LLM-based and BERT-based baselines.”
> >
>
> **Responses:**
> **The current state-of-the-art methods in this domain are indeed BERT-based models [3, 6]**. Therefore, it is necessary for us to compare against all strong BERT-based SOTA baselines. To further ensure fairness, we additionally implemented and integrated a wide range of OOD-detection techniques on top of LLMs, so that LLM-based baselines are evaluated as thoroughly and competitively as possible.
>
> Across all these settings, our model consistently outperforms both BERT-based SOTA methods and enhanced LLM-based baselines by a significant margin, demonstrating the strength of the proposed formulation.
>
> > **W3:**“Lack of evidence for consistency across model backbones.”
> >
>
> **Responses:**
>
> To further address the concern regarding backbone consistency, we additionally conduct baseline experiments on Qwen2.5-7B, complementing our original results based on LLaMA backbones. The complete results are provided in the table above. As shown, UnLLM consistently outperforms EnergyBased, VOS, and NPO (SOTA OOD detection methods) across all ratios (0.25 / 0.5 / 0.75) and across all evaluation metrics (ACC, F1, K-F1, N-F1).
>
> *Table 4: Performance of Qwen2.5-7B based UnLLM on CLINC dataset*
>
> | Backbone | Ratio | Method | ACC(%) | F1(%) | K-F1(%) | N-F1(%) |
> |----------|--------|---------|------|------|-------|--------|
> | **Qwen2.5-7B** | 0.25 | EnergyBased | 85.24 | 78.86 | 78.57 | 89.42 |
> | |  | VOS | 83.28 | 77.11 | 76.82 | 87.85 |
> | |  | NPO | 86.43 | 72.22 | 71.71 | 91.09 |
> | | | **UnLLM** | **87.82** | **80.59** | **80.29** | **91.88** |
> | | 0.50 | EnergyBased | 87.39 | 87.65 | 87.66 | 87.54 |
> | | | VOS | 88.68 | 89.41 | 89.42 | 88.67 |
> | | | NPO | 82.80 | 77.77 | 77.67 | 85.21 |
> | | | **UnLLM** | **88.80** | **93.88** | **93.96** | **87.76** |
> | | 0.75 | EnergyBased | 88.37 | 91.36 | 91.46 | 80.19 |
> | | | VOS | 84.89 | 88.00 | 88.11 | 75.88 |
> | | | NPO | 74.18 | 76.93 | 77.02 | 66.10 |
> | | | **UnLLM** | **94.44** | **96.37** | **96.42** | **90.54** |

---

> ### Author Response · Authors · 2025-11-19
> **Response to Reviewer 163y[6/6]**
>
> > **W4:**“Unclear and unsupported motivation, both theoretically and empirically. Several key statements, such as 'LLMs often exhibit overconfidence in their predictions' (Section 4.3) and 'we observe a misalignment between the LLM’s internal knowledge (representation space) and its outputs' (Section 4.2), are presented as empirical observations but are not supported by any systematic study or quantitative evidence.”
> >
> **Responses:**
>
> We clarify that both “overconfidence” and “representation–output misalignment” are well-established phenomena and are quantitatively verified in our OSTC setting.
>
> **1. Evidence for Overconfidence**:
>
> We add a analysis to substantiate the claim that “LLMs often exhibit overconfidence in their predictions, especially for texts with high semantic similarity to known labels.”
>
> Specifically, for each test sample, we compute its similarity with all candidate labels and record the similarity associated with the predicted label as a measure of its “confusability.” By applying different similarity thresholds, we retain subsets of test samples that are increasingly difficult—that is, samples whose high label similarity typically leads to worse performance under traditional Energy-Based methods but better performance under our UnLLM framework.
>
> The results are shown in the following table. These results directly verify both the existence of the overconfidence problem and the effectiveness of our method in addressing it.
>
> *Table 5: Performance of EnergyBased and UnLLM methods under different similarity thresholds on the CLINC dataset (0.25 KCR).*
>
> | Threshold | Method       |   ACC(%)  |   F1(%)   |  K-F1(%) |  N-F1(%) |
> |-----------|--------------|--------|--------|--------|--------|
> | 0.96      | EnergyBased  | 84.47  | 67.78  | 66.63  | 89.45  |
> |           | **UnLLM**        | **91.93**  | **80.85**  | **80.14**  | **94.42**  |
> | 0.97      | EnergyBased  | 76.92  | 53.65  | 49.21  | 84.75  |
> |           | **UnLLM**        | **97.44**  | **94.19**  | **93.33**  | **98.46**  |
>
> **2. Evidence for Representation–output misalignment**
>
> Li et al. (2023)[5] has demonstrated that LLMs can internally encode correct distinctions while the output layer fails to express them. To validate this in our setting, we compare four model variants, measuring AUC (representation separability, internal recognition) versus K-F1 / N-F1 (output correctness) as shown in Table 3.
>
> Key Findings.
> (1) **Closed-set fine-tuning shows a large gap between high AUC and near-zero N-F1**, indicating that the model internally separates ID/OOD but the output layer cannot express “OOD”—clear evidence of misalignment caused by the closed-set training paradigm.
> For example, at 0.25 ratio, AUC remains 0.91, yet N-F1 is 0.0, and similarly at 0.75 ratio, AUC stays 0.92 while N-F1 is still 0.0.
>
> (2) **Open-set fine-tuning improves both separability and expressiveness.**
> Compared to the LLM-OOD variant, open-set SFT increases N-F1 from 0.68 → 0.87 (0.25 ratio) and from 0.61 → 0.85 (0.75 ratio), alongside AUC improvements (e.g., 0.91 → 0.94 at 0.25).
>
> (3) **Calibration further aligns the output layer, providing consistent boosts across all ratios.**
> For example, at 0.25 ratio, N-F1 improves from 0.87 → 0.89, with AUC also increasing (e.g., 0.94 → 0.96).
>
> **Refs:**
>
> [1] Weitang Liu, Xiaoyun Wang, John Owens, and Yixuan Li. Energy-based out-of-distribution detection. Advances in neural information processing systems, 33:21464–21475, 2020.
> [2] Chuan Guo, Geoff Pleiss, Yu Sun, Kilian Q. Weinberger. On Calibration of Modern Neural Networks. ICML, 2019.
> [3] Yunhua Zhou, Jianqiang Yang, Pengyu Wang, and Xipeng Qiu. Two birds one stone: Dynamic ensemble for OOD intent classification. Association for Computational Linguistics, 2023.
> [4] Dan Hendrycks, Mantas Mazeika, Thomas Dietterich. Deep Anomaly Detection with Outlier Exposure. ICLR, 2019.
> [5] Li, Kenneth, et al. "Inference-time intervention: Eliciting truthful answers from a language model." Advances in Neural Information Processing Systems 36 (2023): 41451-41530.
> [6] Yunhua Zhou, Peiju Liu, and Xipeng Qiu. KNN-contrastive learning for out-of-domain intent classification. Association for Computational Linguistics, 2022.
> [7] Du, Xuefeng, et al. "Vos: Learning what you don't know by virtual outlier synthesis." ICLR, 2022.

---

> ### Comment · Reviewer_163y · 2025-11-23
> **Response to Authors**
>
> Thank you for the patient and detailed rebuttal. I am satisfied with the authors’ rebuttal. The additional experiments and analyses they provided are valuable and help clarify the main findings. These ablations are essential for supporting the proposed claims, and I recommend that they be included in the revised version with clear descriptions. I have adjusted my score accordingly to reflect these improvements.

---

> > ### Author Response · Authors · 2025-11-24
> >
> > We appreciate that we were able to address your concerns, and your recognition means a great deal to us. We will incorporate the additional experiments and corresponding clarifications into the revised version to further strengthen the clarity and completeness of the paper. Thank you again for your careful review and valuable feedback.

---

### Official Review · Reviewer_EwEz · 2025-11-01

**Soundness:** 2
**Presentation:** 1
**Contribution:** 2
**Rating:** 4
**Confidence:** 3

**Summary:**

This paper proposes UnLLM, an unknown-aware large language model designed for open-set text classification. The authors introduce a 3-fold method, including:
1) open-set generative fine-tuning to develop a K+1 classifier.
2) OOD parameter calibration method to align the model’s internal cognitiion of the unknown with its outputs.
3) analogy-augmented self-reflection mechanism to mitigate overconfidence.

Experimental results across six datasets demonstrate that UnLLM consistently achieves SOTA performance in OOD detection while maintaining competitive accuracy in ID classification.

**Strengths:**

* This paper targets an important problem (Open-Set Text Classification), and the proposed method is novel.

* Experimental results across six datasets demonstrate that UnLLM consistently achieves SOTA performance in OOD detection while maintaining competitive accuracy in ID classification.

**Weaknesses:**

1. The writing is not good enough, I find it hard to follow the insights of the proposed method and the details of the equations. For example,
* What are the insights behind adding contrastive learning loss?
* In Equation 2, the authors say $N_{y_{i,j}$ is the number of examples of $y_{i,j}$, but what happens when the model is optimized in batch?
* Why authors could "assume that the representation for each class k ∈ [1,K] follows a class-conditional
Gaussian distribution:"?

2. I have a concern about section `Calibration Direction Construction`, where authors propose to "evaluate the fine-tuned LLM on the label partitioned validation set". I don't think using statistics of validation set is a fair comparison to other baselines.

3. The proposed 3-fold methods seem to have no relation, which makes me feel that LLM is too complicated.

**Questions:**

Please refer to Weaknesses.

---

> ### Author Response · Authors · 2025-11-19
> **Response to Reviewer EwEz[1/2]**
>
> Thank you for your valuable comments. We respond to each weakness in detail below.
>
> > **W1:**“What are the insights behind adding contrastive learning loss?”
> >
>
> **Responses:** Added to Representation Modeling (Sec. 4.1) to ensure compact intra-class embeddings and **separated inter-class/OOD representations**. This prevents the LLM from collapsing semantically close OOD and ID samples into the same cluster. Intuitively, contrastive learning regularizes feature geometry to create **clear open-set boundaries**, which is vital for OOD awareness.
>
> > **W2:**“In Equation 2, the authors say $N_{y_{i,j}y_{i,j}}$, but what happens when the model is optimized in batch?”
> >
>
> **Responses:** The expression $N_{y_{i,j}}$ represents the number of samples of class $y_{i,j}$ within a mini-batch. During batch training, positive pairs are drawn from samples in the same batch belonging to the same class, and negatives are all others. This follows the standard normalized temperature-scaled framework [1].
>
>
> > **W3:**“Why authors could "assume that the representation for each class k ∈ [1,K] follows a class-conditional Gaussian distribution:"?”
> >
>
> **Responses:**
> This assumption is a statistical approximation, previously adopted in [2] to estimate each class’s mean and variance in the embedding space.
>
> > **W4:**“I have a concern about section Calibration Direction Construction, where authors propose to "evaluate the fine-tuned LLM on the label partitioned validation set". I don't think using statistics of validation set is a fair comparison to other baselines.”
> >
>
> **Responses:**
>
> Prior SOTA methods such as KnnCon [3] and DyEn [4]—both of which we include as baselines—our approach likewise relies on the validation set to determine the LOF threshold. Conceptually, our method follows the same principle: we use the validation set only to estimate the parameters needed for calibrating the classification weight of the single OOD token.

---

> ### Author Response · Authors · 2025-11-19
> **Response to Reviewer EwEz[2/2]**
>
> > **W5:**“The proposed 3-fold methods seem to have no relation, which makes me feel that LLM is too complicated.”
> >
>
> **Responses:**
>
> We would like to clarify that the three components of our approach are **not independent tricks**, but are designed as **mutually reinforcing elements within a unified LLM-centric open-set framework**. Our method is grounded in a novel formulation that dynamically samples label subsets to construct pseudo-OOD examples, enabling a K+1 generative fine-tuning paradigm tailored for LLMs.
>
> Within this framework:
> - LLMs are enabled to perform both OOD detection and accurate classification of ID samples without manual thresholding, making the approach particularly well-suited for open-set text classification.
> - Contrastive learning and orthogonality are used **not as standalone methods**, but as **integrated representation-level constraints** to improve class separation to enforce the model to rely on ID-specific independent features while suppressing redundant, OOD-correlated information.
> - The use of pseudo-OOD samples also facilitates **calibration**, where parameter-level alignment is introduced to better associate internal activations with unknown semantics.
> - **Reflective inference** is effective within our K+1 formulation, because the model learns an explicit ID–OOD semantic boundary that makes reflection meaningful—unlike ID-only LLMs, where lacking an OOD prior leads to unstable or negligible gains.
>
> In summary, our approach establishes a **new LLM-centric paradigm** for open-set text classification, which is fundamentally different from existing discriminative methods.
>
> To substantiate this point, we conducted an additional ablation study in which we replaced the Open-set SFT with Closed-set SFT, and sequentially integrated each module into both the Closed-set and Open-set pipelines for comparison. The results are summarized in the table below.
>
> The empirical results show that when modules are added on top of Closed-set SFT, the model fails to mitigate large-scale overfitting of LLM outputs to the ID label space. In fact, performance drops from an F1 of 0.48 to 0.44 (**a relative degradation of 9.48%**). Even with LLM-OOD post-hoc detection, the improvement is only marginal, reaching 0.73 F1, and still cannot overcome the inherent bottleneck.
>
> In contrast, Open-set SFT alone already yields a substantial improvement, increasing F1 from 0.48 to 0.75 (**a relative gain of 55.99%**).
> Furthermore, only when the subsequent modules are applied on top of Open-set SFT does the model reach its best performance of 0.83 F1 (**a relative gain of 90.09% and 13.32% compared with Closed-set SFT and LLM-OOD equipped with the same modules**).
>
> *Table 1: Ablation Study on CLINC dataset of 0.25 ratio.*
>
> | Method                       || ACC(%)            | F1(%)       | K-F1(%)          | N-F1(%)                |
> |------------------------------|----------------|----------------|----------|---------------|---------------------|
> |LLM Output| Closed-set SFT                           | 24.31          | 48.40    | 49.71         | 0.00                |
> || Closed-set SFT  + Contrastive Learning   | 24.53          | 48.50    | 49.80         | 0.35                |
> || Closed-set SFT  + Contrastive Learning + Orthogonal Regularizer | 24.58          | 47.53    | 48.79         | 0.82                |
> || Closed-set SFT  + Contrastive Learning + Orthogonal Regularizer + Analogy                | 36.89          | 43.81    | 43.98         | 37.27               |
> |Post-hoc Method| LLM-OOD  |   65.66  |   66.68    |       66.65  | 67.89      |
> || LLM-OOD + Contrastive Learning  |  68.00   |  30.21     |   29.71      |   78.26    |
> || LLM-OOD + Contrastive Learning + Orthogonal Regularizer  |  81.02   |  74.33     |   73.68      |    86.28    |
> || LLM-OOD + Contrastive Learning + Orthogonal Regularizer + Analogy  |  80.8   |  73.49     |   73.15      |   86.16    |
> |LLM Output| Open-set SFT  |   80.18|	75.5|	75.24|	85.3|
> || Open-set SFT  + Contrastive Learning |  82.22|	77.57|	77.32|	86.88|
> || Open-set SFT  + Contrastive Learning + Orthogonal Regularizer  |83.51|	78.98|	78.73|	88.09|
> || Open-set SFT  + Contrastive Learning + Orthogonal Regularizer + Calibaration   |85.09|	80.32|	80.09|	90.04|
> || Open-set SFT  + Contrastive Learning + Orthogonal Regularizer + Calibaration + Analogy |**89.87** |**83.28**| **83.01**| **93.15**|
>
> **Refs:**
>
> [1] Chen, Ting, et al. "A simple framework for contrastive learning of visual representations." International conference on machine learning. PmLR, 2020.
> [2] Du, Xuefeng, et al. "Vos: Learning what you don't know by virtual outlier synthesis." ICLR (2022).
> [3] Yunhua Zhou, Peiju Liu, and Xipeng Qiu. 2022. KNN-Contrastive Learning for Out-of-Domain Intent Classification. ACL 2022.
> [4] Yunhua Zhou, Jianqiang Yang, Pengyu Wang, and Xipeng Qiu. 2023. Two Birds One Stone: Dynamic Ensemble for OOD Intent Classification. ACL 2023.

---

> ### Author Response · Authors · 2025-11-26
> **Kind Reminder for Your Response**
>
> Dear Reviewer EwEz,
>
> I hope this message finds you well. As the discussion period is nearing its end, with less than one week remaining, I wanted to ensure that we have addressed all of your concerns satisfactorily. If there are any additional points or feedback you would like us to consider, please let us know. Your insights are invaluable to us, and we are eager to address any remaining issues to improve our work.
>
> Thank you for your time and effort in reviewing our paper.

---

### Author Response · Authors · 2025-12-02

Dear AC,

We sincerely appreciate the time and effort that you and the reviewers have devoted to evaluating our submission and considering our rebuttal. We are also grateful that the reviewers recognize the value and potential impact of our work.

**We respectfully ask the AC to consider the  post-rebuttal scores prior to the information leak—6 (Reviewer rWfQ), 4 (Reviewer EwEz), 6 (Reviewer 163y)—along with the fact that all concerns raised by Reviewer EwEz have been thoroughly addressed, although they have not responded.**

Overall, the majority of reviewers provided positive feedback on both our methodology and empirical contributions, highlighting the **novelty, coherence, and practicality** of our proposed three-stage approach.

On the methodological side:
>**Reviewer rWfQ**: The three-stage pipeline (contrastive + orthogonality regularization; K+1 logit calibration; analogy-augmented inference) is **coherent and practical**.
>**Reviewer EwEz**: This paper targets an important problem (Open-Set Text Classification), and the proposed method is **novel**.
>**Reviewer 163y**: This framing is **new** in the context of large language models and **conceptually straightforward**.

On the experimental side:
>**Reviewer rWfQ**: Shows **consistent improvements** on several benchmarks.
>**Reviewer EwEz**: Experimental results across **six** datasets demonstrate that UnLLM consistently achieves **SOTA performance** in OOD detection while maintaining competitive accuracy in ID classification.
>**Reviewer 163y**: The authors evaluate their approach on **six** benchmark datasets and a broad range of baselines, covering both traditional PLM-based and more recent LLM-based methods. The experimental coverage is thorough and shows **consistent empirical improvements**.

During the rebuttal phase, Reviewer 163y expressed clear satisfaction with our responses and explicitly raised their score from **2 to 6**:
>**Reviewer 163y**: I am satisfied with the authors’ rebuttal. **I have adjusted my score accordingly to reflect these improvements**.

At this point, the only reviewer who has not responded—and the only one with a remaining negative score—is Reviewer EwEz. We have addressed all of their concerns comprehensively, and we hope the AC will take this into consideration.

Thank you again for your time, your support, and your careful evaluation of our work.

---

### Meta-Review · Area_Chair_ynUc · 2026-01-06

**Summary:**

Across reviewers, the primary concerns centered on clarity, scientific depth, and disentangling the contributions of the multiple components in UnLLM, rather than on empirical effectiveness. Reviewers EwEz and 163y both noted that the paper is difficult to follow, with insufficiently explained motivations, assumptions, and equations, and raised concerns that the three-fold framework (subset-conditioned generation, representation regularization, calibration, and reflective inference) appears loosely connected. Reviewer 163y in particular questioned whether the proposed subset-conditioned reformulation is essential, arguing that much of the performance gain may stem from well-known components such as contrastive learning, orthogonality constraints, or inference-time reflection, which could plausibly be attached to existing methods. Additional concerns included potentially unfair comparisons between large LLMs and much smaller PLM baselines, limited analysis of why existing methods fail, unclear theoretical justification for some design choices (e.g., Gaussian assumptions, orthogonality objectives), and presentation issues that obscure the core insights. These concerns led to divergent initial scores, with skepticism driven mainly by lack of explanatory depth rather than by doubts about correctness.

**Reviewer Concerns:**

### Concerns addressed by the rebuttal

Reviewer EwEz

- Poor clarity around key design choices (contrastive loss, batch formulation, Gaussian assumption): Largely addressed. The rebuttal gives concrete intuition for contrastive learning in open-set geometry, clarifies the mini-batch interpretation of Eq. 2, and justifies the Gaussian approximation via precedent (VOS).

- Fairness of using validation statistics in calibration: Addressed. The rebuttal argues parity with common practice in included baselines (e.g., KNNCon, DyEn) and reframes the validation usage as parameter/threshold estimation rather than extra supervision.

- “Three-fold methods are unrelated / too complicated”: Addressed with a more coherent systems-level story and, importantly, a new control ablation showing modules help only when built on open-set SFT.

Reviewer 163y

- Essentiality of the subset-conditioned reformulation: Substantially addressed. The rebuttal adds the exact control experiment requested: replacing open-set SFT with closed-set SFT and layering the same modules on both; results indicate the reformulation is the key enabling factor.

- Modules could be applied to baselines; gains may come from known tricks: Partially-to-strongly addressed. The rebuttal provides evidence that CL/orth/analogy behave very differently under closed-set vs open-set training and argues the orthogonality objective is a novel training loss (not just a post-hoc score).

- Reflective inference being independent: Addressed with evidence. The rebuttal shows analogy/reflection yields little benefit under LLM-OOD but meaningful gains under open-set SFT, suggesting dependence on an explicit OOD boundary learned in training.

- Lack of evidence for “misalignment” and “overconfidence” claims: Addressed with additional quantitative analyses (AUC vs N-F1 gap; similarity-threshold stress tests) and citation support (Li et al. 2023).

- Backbone generality: Addressed. They added baseline comparisons on Qwen2.5-7B (not just UnLLM), directly responding to the portability concern.

- “Unfair” LLM vs BERT comparisons: Partially addressed. The rebuttal defends the necessity of comparing against BERT SOTA in OSTC and claims stronger LLM baselines were implemented; however, it does not fully resolve the conceptual critique that capacity differences confound conclusions about “LLMs vs PLMs” as a paradigm.

Reviewer rWfQ

- Closeness to OE-style methods (WOODS/VOS/NPOS): Addressed at a positioning level. The rebuttal clarifies “no external OOD data” and “real labeled OOD via subset partitioning” vs virtual/outlier exposure—this helps differentiate the setting and method.

- Analogy module details and latency: Addressed with concrete implementation details (encoder choice, k, index size) and an empirical overhead estimate.

- Requests for conditional-OOD vs cross-domain controlled test / adding auxiliary-OOD methods: Addressed as out-of-scope. The rebuttal clearly states the assumed OSTC setting disallows external OOD data and argues these comparisons are not “fair” under that definition.

### Concerns still outstanding (or only partially resolved)

- Writing quality / presentation depth (EwEz, 163y). The rebuttal improves explanations, but it’s unclear whether the paper text itself has been revised enough to fix the “hard to follow” critique (especially EwEz’s “presentation: poor”). The rebuttal provides clarifications, but acceptance-quality presentation requires integrating them cleanly into the manuscript (equations, assumptions, training/inference pipeline, and definitions).

- Scientific depth / mechanistic understanding (163y).
The new ablations and representation metrics strengthen the empirical argument, but the core critique—limited scientific explanation of why the reformulation works and why prior methods fail—may still be only partially resolved. The rebuttal adds evidence, but not necessarily a deeper causal/mechanistic account beyond “overlap exists; our objective separates.”

- Fairness and framing of model-capacity comparisons (163y).
They defend comparing to BERT because it is SOTA in the domain, but the concern about drawing broad conclusions from LLM-vs-BERT comparisons remains unless the paper explicitly avoids over-claiming and/or adds capacity-matched comparisons or scaling discussion.

**Reviewer Scores:**

- Reviewer EwEz (score: 4): Likely would have increased to 6. The rebuttal directly answers the concrete technical confusions (contrastive loss insight, batch form of Eq. 2, Gaussian assumption) and defends validation-set usage by aligning with baseline practice, plus it adds a strong ablation showing the modules are coherent under open-set SFT. Remaining risk is whether the revised paper text becomes sufficiently clear.

- Reviewer 163y (score: 2): Already expressed the willingness to increase to 6. The rebuttal substantially addresses their core critique by adding the exact control experiments they requested (modules on closed-set vs open-set SFT), plus representation-separation evidence, calibration diagnostics (AUC vs N-F1), and baseline comparisons on Qwen. However, their “limited scientific depth / unclear rationale” concern is only partially resolved, and they may still view the contribution as empirically strong but not conceptually convincing enough for a full accept.

- Reviewer rWfQ (score: 6): Likely remain still. The rebuttal clarifies novelty versus OE-style methods, provides concrete analogy retrieval/latency details, and appropriately scopes out auxiliary-OOD methods. Some requested extras (mean±std, reliability diagrams) aren’t explicitly addressed, so the increase would be modest rather than large.

---

### Decision · Program_Chairs · 2026-01-26

Accept (Poster)